# Encoding Time-Series Explanations through Self-Supervised Model Behavior Consistency

**Owen Queen**
Harvard University
owen_queen@hms.harvard.edu

**Thomas Hartvigsen**
University of Virginia, MIT
hartvigsen@virginia.edu

**Teddy Koker**
MIT Lincoln Laboratory
thomas.koker@ll.mit.edu

**Huan He**
Harvard University
huan_he@hms.harvard.edu

**Theodoros Tsiligkaridis**
MIT Lincoln Laboratory
tsili@ll.mit.edu

**Marinka Zitnik**
Harvard University
marinka@hms.harvard.edu

## Abstract

Interpreting time series models is uniquely challenging because it requires identifying both the location of time series signals that drive model predictions and their matching to an interpretable temporal pattern. While explainers from other modalities can be applied to time series, their inductive biases do not transfer well to the inherently challenging interpretation of time series. We present TIMEX, a time series consistency model for training explainers. TIMEX trains an interpretable surrogate to mimic the behavior of a pretrained time series model. It addresses the issue of model faithfulness by introducing *model behavior consistency*, a novel formulation that preserves relations in the latent space induced by the pretrained model with relations in the latent space induced by TIMEX. TIMEX provides discrete attribution maps and, unlike existing interpretability methods, it learns a latent space of explanations that can be used in various ways, such as to provide landmarks to visually aggregate similar explanations and easily recognize temporal patterns. We evaluate TIMEX on eight synthetic and real-world datasets and compare its performance against state-of-the-art interpretability methods. We also conduct case studies using physiological time series. Quantitative evaluations demonstrate that TIMEX achieves the highest or second-highest performance in every metric compared to baselines across all datasets. Through case studies, we show that the novel components of TIMEX show potential for training faithful, interpretable models that capture the behavior of pretrained time series models.

## 1 Introduction

Prevailing time series models are high-capacity pre-trained neural networks [1, 2], which are often seen as black boxes due to their internal complexity and lack of interpretability [3]. However, practical use requires techniques for auditing and interrogating these models to rationalize their predictions. Interpreting time series models poses a distinct set of challenges due to the need to achieve two goals: pinpointing the specific *location* of time series signals that influence the model's predictions and aligning those signals with *interpretable temporal patterns* [4]. While explainers designed for other modalities can be adapted to time series, their inherent biases can miss important structures in time series, and their reliance on isolated visual interpretability does not translate effectively to

37th Conference on Neural Information Processing Systems (NeurIPS 2023).

the time series where data are less immediately interpretable. The dynamic nature and multi-scale dependencies within time series data require temporal interpretability techniques.

Research in model understanding and interpretability developed *post-hoc* explainers that treat pretrained models as black boxes and do not need access to internal model parameters, activations, and gradients. Recent research, however, shows that such *post-hoc* methods suffer from a lack of faithfulness and stability, among other issues [5, 6, 7]. A model can also be understood by investigating what parts of the input it attends to through attention mapping [8, 9, 10] and measuring the impact of modifying individual computational steps within a model [11, 12]. Another major line of inquiry investigates internal mechanisms by asking what information the model contains [13, 14, 15]. For example, it has been found that even when a language model is conditioned to output falsehoods, it may include a hidden state that represents the true answer internally [16]. The gap between external failure modes and internal states can only be identified by probing model internals. Such representation probing has been used to characterize the behaviors of language models, but leveraging these strategies to understand time series models has yet to be attempted. These lines of inquiry drive the development of *in-hoc* explainers [17, 18, 19, 20, 21, 22] that build inherent interpretability into the model through architectural modifications [18, 19, 23, 20, 21] or regularization [17, 22]. However, no *in-hoc* explainers have been developed for time series data. While explainers designed for other modalities can be adapted to time series, their inherent biases do not translate effectively to the uninterpretable nature of time series data. They can miss important structures in time series.

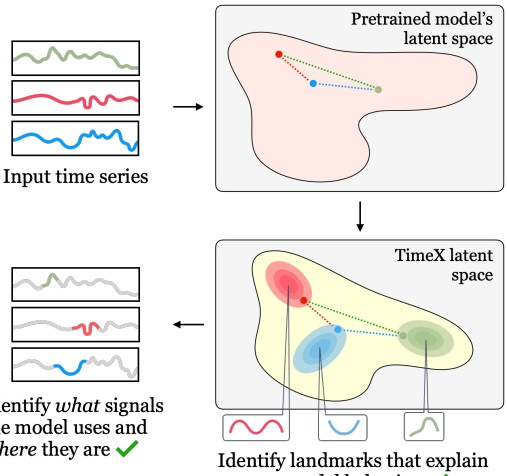

Figure 1: TIMEX learns a latent space of explanations and landmarks to summarize groups of informative temporal patterns in time series.

Explaining time series models is challenging for many reasons. First, unlike imaging or text datasets, large time series data *are not visually interpretable*. Next, time series often exhibit dense informative features, unlike more explored modalities such as imaging, where informative features are often sparse. In time series datasets, timestep-to-timestep transitions can be negligible, and temporal patterns only show up when *looking at time segments and long-term trends*. In contrast, in language datasets, word-to-word transitions are informative for language modeling and understanding. Further, time series interpretability involves understanding the dynamics of the model and identifying trends or patterns. Another critical issue with applying prior methods is that they treat all *time steps as separate features*, ignoring potential time dependencies and contextual information; we need explanations that are *temporally connected* and *visually digestible*. While understanding predictions of individual samples is valuable, the ability to establish *connections between explanations of various samples* (for example, in an appropriate latent space) could help alleviate these challenges.

**Present work.** We present TIMEX, a novel time series surrogate explainer that produces interpretable attribution masks as explanations over time series inputs (Figure 1). ① A key contribution of TIMEX is the introduction of *model behavior consistency,* a novel formulation that ensures the preservation of relationships in the latent space induced by the pretrained model, as well as the latent space induced by TIMEX. ② In addition to achieving model behavior consistency, TIMEX offers interpretable attribution maps, which are valuable tools for interpreting the model's predictions, generated using discrete straight-through estimators (STEs), a type of gradient estimator that enables end-to-end training of TIMEX models. ③ Unlike existing interpretability methods, TIMEX goes further by learning a latent space of explanations. By incorporating model behavior consistency and leveraging a latent space of explanations, TIMEX provides discrete attribution maps and enables visual aggregation of similar explanations and the recognition of temporal patterns. ④ We test our approach on eight synthetic and real-world time series datasets, including datasets with carefully processed ground-truth explanations to quantitatively benchmark it and compare it to general explainers, state-of-the-art time series explainers, and *in-hoc* explainers. TIMEX is at `https://github.com/mims-harvard/TimeX`

## 2 Related work

**Model understanding and interpretability.** As deep learning models grow in size and complexity, so does the need to help users understand the model's behavior. The vast majority of explainable AI research (XAI) [24] has focused on natural language processing (NLP) [25, 26, 27] and computer vision (CV) [28, 29, 30]. Commonly used techniques, such as Integrated Gradients [31] and Shapley Additive Explanations (SHAP) [3], and their variants have originated from these domains and gained popularity. XAI has gained significant interest in NLP and CV due to the inherent interpretability of the data. However, this familiarity can introduce confirmation bias [32]. Recent research has expanded to other data modalities, including graphs [33, 6] and time series [34, 35], as outlined below. The literature primarily focuses on *post-hoc* explainability, where explanations are provided for a trained and frozen model's behavior [36, 37]. However, saliency maps, a popular approach [38, 31, 39], have pitfalls when generated *post-hoc*: they are surprisingly fragile [5], and lack sensitivity to their explained models [40]. Surrogate-based approaches have also been proposed [41, 42, 43], but these simplified surrogate models fall short compared to the original predictor they aim to explain. Unlike *post-hoc* explainability, *in-hoc* methods aim for inherently interpretable models. This can be accomplished by modifying the model's architecture [20], training procedure using jointly-trained explainers [44], adversarial training [45, 46, 47, 48], regularization techniques [17, 22], or refactorization of the latent space [49, 50]. However, such models often struggle to achieve state-of-the-art predictive performance, and to date, these methods have seen limited use for time series.

**Beyond instance-based explanations.** Several methods have been proposed to provide users with information on model behavior beyond generating instance-based saliency maps explaining individual predictions. Prototype models strive to offer a representative sample or region in the latent space [51, 52]. Such methods are inherently interpretable, as predictions are directly tied to patterns in the feature space. Further, explainability through human-interpretable exemplars has been gaining popularity. Concept-based methods decompose model predictions into human-interpretable *concepts*. Many works rely on annotated datasets with hand-picked concepts (*e.g.*, "stripes" in an image of a zebra). Relying on access to *a priori* defined concepts, concept bottleneck models learn a layer that attributes each neuron to one concept [23]. This limitation has spurred research in concept discovery by composing existing concepts [53, 54] or grounding detected objects to natural language [55]. However, the CV focus of these approaches limits their applicability to other domains like time series.

**Time series explainability.** In contrast to other modalities, time series often have multiple variables, and their discriminative information is spread over many timesteps. Building on these challenges, recent works have begun exploring XAI for time series [56, 57, 58, 59, 60, 61, 34, 62]. Many methods modify saliency maps [35, 63, 58] or surrogate methods [59, 64] to work with time series data. Two representative methods are WinIT [65] and Dynamask [58]. WinIT learns saliency maps with temporal feature importances, while Dynamask regularizes saliency maps to include temporal smoothing. However, these methods rely on perturbing timesteps [63], causing them to lack faithfulness. Common perturbation choices in CV, like masking with zeros, make less sense for time series [56]. Perturbed time series may be out-of-distribution for the model due to shifts in shape [66], resulting in unfaithful explanations akin to adversarial perturbation [67].

## 3 Problem formulation

**Notation.** Given is a time series dataset $\mathcal{D} = (\mathcal{X}, \mathcal{Y}) = \{(\mathbf{x}_i, y_i) | i = 1, ..., N\}$ where $\mathbf{x}_i$ are input samples and $y_i$ are labels associated to each sample. Each sample $\mathbf{x}_i \in \mathbb{R}^{T \times d}$ is said to have $T$ time steps and $d$ sensors. A *feature* is defined as a time-sensor pair, where the time $t$ and sensor $k$ for input $\mathbf{x}_i$ is $\mathbf{x}_i[t, k]$. Without loss of generality, we consider univariate ($d = 1$) and multivariate ($d > 1$) settings. Each $y_i \in \{1, 2, ..., C\}$ belongs to one of $C$ classes. A classifier model consists of an encoder $G$ and predictor $F$. The encoder $G$ produces an embedding of input $\mathbf{x}_i$, *i.e.*, $G(\mathbf{x}_i) = \mathbf{z}_i \in \mathbb{R}^{d_z}$, while the predictor produces some prediction from the embedding in the form of a logit, *i.e.*, $F(G(\mathbf{x}_i)) = \hat{y}_i \in [0, 1]^C$ where $\text{argmax}_j \hat{y}_i[j] \in \{1, ..., C\}$ is the predicted label.

The latent space induced by $G$ is defined as $Z$, *e.g.*, $G : \mathcal{X} \to Z$. We will refer to $F(G(\cdot))$ as the reference model while $G$ is the *reference encoder* and $F$ is the *reference predictor*. An explanation is defined as a continuous map of the features that conveys the relative importance of each feature for the prediction. The explanation for sample $\mathbf{x}_i$ is given as an attribution map $E(\mathbf{x}_i) \in \mathbb{R}^{T \times d}$ where for any times $t_1, t_2$ and sensors $k_1, k_2$, $E(\mathbf{x}_i[t_1, k_1]) > E(\mathbf{x}_i[t_2, k_2])$ implies that $\mathbf{x}_i[t_1, k_1]$ is a more

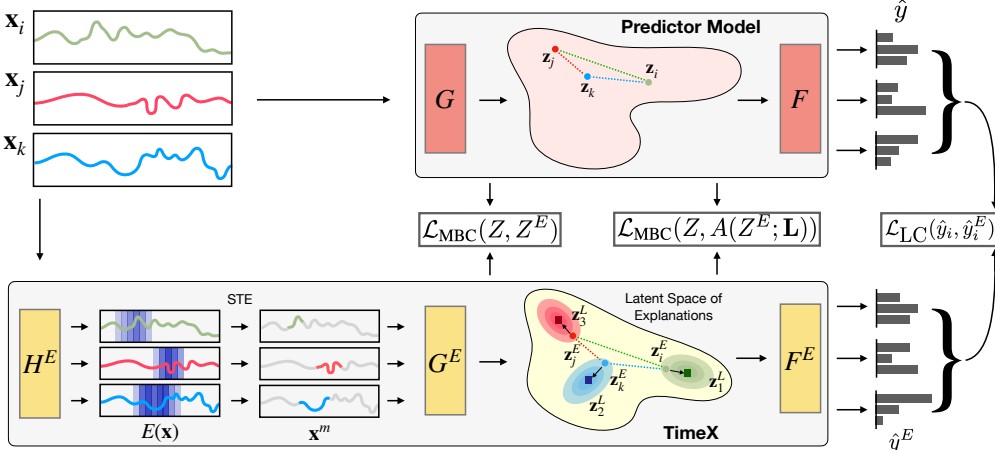

Figure 2: Overview of TIMEX approach.

important feature for the task than $\mathbf{x}_i[t_2, k_2]$. Finally, we define an occlusion procedure whereby a function $\Omega$ generates an mask $M_{\mathbf{x}}$ for a sample $\mathbf{x}$ from the explanation $E(\mathbf{x})$, e.g., $\Omega(E(\mathbf{x})) = M_{\mathbf{x}}$. This mask is applied to $x$ to derive a masked version of the input $\mathbf{x}^m$ through an operation $\odot$, e.g., $M_{\mathbf{x}} \odot \mathbf{x} = \mathbf{x}^m$. When describing TIMEX, we generally refer to $\odot$ as an element-wise multiplication.

## 3.1 Self-supervised model behavior consistency

TIMEX creates an inherently-interpretable surrogate model for pretrained time series models. The surrogate model produces explanations by optimizing two main objectives: interpretability and faithfulness to model behavior. First, TIMEX generates interpretable explanations via an attribution map $E(\mathbf{x}_i)$ that identifies succinct, connected regions of input important for the prediction. To ensure faithfulness to the reference model, we introduce a novel objective for training TIMEX: *model behavior consistency (MBC)*. With MBC, a TIMEX model learns to mimic internal layers and predictions of the reference model, yielding a high-fidelity time series explainer. MBC is defined as:

**Definition 3.1** (Model Behavior Consistency (MBC)). Explanation $E$ and explanation encoder $G^E$ are consistent with pretrained model $G$ and predictor $F$ on dataset $\mathcal{D}$ if the following is satisfied:

- **Consistent reference encoder:** Relationship between $\mathbf{z}_i = G(\mathbf{x}_i)$ and $\mathbf{z}_j = G(\mathbf{x}_j)$ in the space of reference encoder is preserved by the explainer, $\mathbf{z}_i^E = G^E(\mathbf{x}_i^m)$ and $\mathbf{z}_j^E = G^E(\mathbf{x}_j^m)$, where $\mathbf{x}_i^m = \Omega(E(\mathbf{x}_i)) \odot \mathbf{x}_i$ and $\mathbf{x}_j^m = \Omega(E(\mathbf{x}_j)) \odot \mathbf{x}_j$, through distance functions on the reference encoder's and explainer's latent spaces $D_Z$ and $D_{Z^E}$, respectively, such that: $D_Z(\mathbf{z}_i, \mathbf{z}_j) \simeq D_{Z^E}(\mathbf{z}_i^E, \mathbf{z}_j^E)$ for samples $\mathbf{x}_i, \mathbf{x}_j \in \mathcal{D}$.
- **Consistent reference predictor:** Relationship between reference predictor $\hat{y}_i = F(\mathbf{z}_i)$ and latent explanation predictor $\hat{y}_i^E = F^E(\mathbf{z}_i^E)$ is preserved, $\hat{y}_i \simeq \hat{y}_i^E$ for every sample $\mathbf{x}_i \in \mathcal{D}$.

Our central formulation is defined as realizing the MBC between a reference model and an interpretable TIMEX model:

**Problem statement 3.1** (TIMEX). Given pretrained time series encoder $G$ and predictor $F$ that are trained on a time series dataset $\mathcal{D}$, TIMEX provides explanations $E(\mathbf{x}_i)$ for every sample $\mathbf{x}_i \in \mathcal{D}$ in the form of interpretable attribution maps. These explanations satisfy *model behavior consistency* through the latent space of explanations $Z^E$ generated by the explanation encoder $G^E$.

TIMEX is designed to counter several challenges in interpreting time series models. First, TIMEX avoids the pitfall known as the *occlusion problem* [68]. Occlusion occurs when some features in an input $\mathbf{x}_i$ are perturbed in an effort that the predictor forgets those features. Since it is well-known that occlusion can produce out-of-distribution samples [69], this can cause unpredictable shifts in the behavior of a fixed, pretrained model [70, 71, 72]. In contrast, TIMEX avoids directly masking input samples to $G$. TIMEX trains an interpretable surrogate $G^E$ to match the behavior of $G$. Second, MBC is designed to improve the faithfulness of TIMEX to $G$. By learning to mimic multiple states of $F(G(\cdot))$ using the MBC objective, TIMEX learns highly-faithful explanations, unlike many *post hoc* explainers that provide no explicit optimization of faithfulness. Finally, TIMEX's explanations are driven by learning a latent explanation space, offering richer interpretability data.

## 4 TIMEX method

We now present TIMEX, an approach to train an interpretable surrogate model to provide explanations for a pretrained time series model. TIMEX learns explanations through a consistency learning objective where an explanation generator $H^E$ and explanation encoder $G^E$ are trained to match intermediate feature spaces and the predicted label space. We will break down TIMEX in the following sections by components: $H^E$, the explanation generator, $G^E$, the explanation encoder, and the training objective of $G^E(H^E(\cdot))$, followed by a discussion of practical considerations of TIMEX. An overview of TIMEX is depicted in Figure 2.

### 4.1 Explanation generation

Generating an explanation involves producing an explanation map $E(\mathbf{x})$ where if $E(\mathbf{x}[t_1, k_1]) > E(\mathbf{x}[t_2, k_2])$, then feature $\mathbf{x}[t_1, k_1]$ is considered as more important for the prediction than $\mathbf{x}[t_2, k_2]$. Explanation generation is performed through an explanation generator $H^E : \mathcal{X} \to [0,1]^{T \times d}$, where $H^E(\mathbf{x}) = \mathbf{p}$. We learn $\mathbf{p}$ based on a procedure proposed by [50], but we adapt their procedure for time series. Intuitively, $\mathbf{p}$ parameterizes a Bernoulli at each time-sensor pair, and the mask $M_{\mathcal{X}}$ is sampled from this Bernoulli distribution during training, *i.e.*, $M_{\mathcal{X}} \sim \mathbb{P}_{\mathbf{p}}(M_{\mathcal{X}}|\mathcal{X}) = \prod_{t,k} \text{Bern}(\mathbf{p}_{t,k})$. This parameterization is directly interpretable as attribution scores: a low $\mathbf{p}_{t,k}$ means that time-sensor pair $(t, k)$ has a low probability of being masked-in. Thus, $\mathbf{p}$ is also the explanation for $\mathbf{x}_i$, *i.e.*, $E(\mathbf{x}) = \mathbf{p}$.

The generation of $\mathbf{p}$ is regularized through a divergence with Bernoulli distributions $\text{Bern}(r)$, where $r$ is a user-chosen hyperparameter. As in [50], denote the desired distribution of $\mathbf{p}$ as $\mathbb{Q}(M_{\mathcal{X}}) = \prod_{(t,k)} \text{Bern}(r)$. Then the objective becomes:

$$\mathcal{L}_m(\mathbf{p}) = \mathbb{E}[D_{\text{KL}}(\mathbb{P}_{\mathbf{p}}(M_{\mathcal{X}}|\mathcal{X})||\mathbb{Q}(M_{\mathcal{X}}))] = \sum_{t,k} \mathbf{p}_{t,k} \log \frac{\mathbf{p}_{t,k}}{r} + (1 - \mathbf{p}_{t,k}) \log \frac{1 - \mathbf{p}_{t,k}}{1 - r} \quad (1)$$

The sampling of $M_{\mathcal{X}} \sim \mathbb{P}_{\mathbf{p}}(M_{\mathcal{X}}|\mathcal{X})$ is performed via the Gumbel-Softmax trick [73, 74], which is a differentiable approximation of categorical sampling. Importantly, $M_{\mathcal{X}}$ is stochastically generated, which, as discussed in [50, 75], regularizes the model to learn robust explanations.

To generate interpretable attribution masks, TIMEX optimizes for the connectedness of predicted distributions:

$$\mathcal{L}_{\text{con}}(\mathbf{p}) = \frac{1}{T \times d} \sum_{k=1}^{d} \sum_{t=1}^{T-1} \sqrt{(\mathbf{p}_{t,k} - \mathbf{p}_{t+1,k})^2}. \quad (2)$$

The generator of explanations $H^E$ learns directly on input time series samples $\mathcal{X}$ to return $\mathbf{p}$. We build a transformer encoder-decoder structure for $H^E$, using an autoregressive transformer decoder and a sigmoid activation to output probabilities for each time-sensor pair.

### 4.2 Explanation encoding

We now describe how to embed explanations with the explanation encoder $G^E$. Intuitively, $G^E$ learns on the masked distribution of $\mathcal{X}$, which can be denoted as $\mathcal{X}^m$. Motivated by the occlusion problem, we avoid directly applying the masks onto the pretrained, frozen $G$, as $\mathcal{X}^m$ and $\mathcal{X}$ are fundamentally different distributions. Therefore, we copy the weights of $G$ into $G^E$ and fine-tune $G^E$ on $\mathcal{X}^m$.

**Discretizing attribution masks.** When passing inputs to $G^E$, it is important for the end-to-end optimization to completely ignore regions identified as unimportant by $H^E$. Therefore, we use a straight-through estimator (STE) [73] to obtain a discrete mask $M_{\mathcal{X}} \in \{0, 1\}^{T \times d}$. Introduced by [76], STEs utilize a surrogate function to approximate the gradient of a non-differentiable operation used in the forward pass, such as binary thresholding.

**Applying masks to time series samples.** We use two types of masking procedures: attention masking and direct-value masking. First, we employ differentiable attention masking through a multiplicative operation proposed by Nguyen et al. [77]. When attention masking does not apply, we use a direct-value masking procedure based on architecture choice or multivariate inputs. We approximate a baseline distribution: $\mathbb{B}_{\mathcal{X}} = \prod_{t,k} \mathcal{N}(\mu_{tk}, \sigma_{tk}^2)$, where $\mu_{tk}$ and $\sigma_{tk}^2$ are the mean and

variance over time-sensor pairs. Masking is then performed through a multiplicative replacement as: $\mathbf{x}_i^m = (M_{\mathcal{X}} \odot \mathbf{x}_i) + (1 - M_{\mathcal{X}}) \odot b$, where $b \sim \mathbb{B}_{\mathcal{X}}$.

**Justification for discrete masking.** It is essential that masks $M_{\mathcal{X}}$ are discrete instead of continuous. Previous works have considered masking techniques [78, 49, 50] with continuous masks since applying such masks is differentiable with element-wise multiplication. However, continuous masking has a distinctly different interpretation: it uses a continuous deformation of the input towards a baseline value. While such an approach is reasonable for data modalities with discrete structures, such as sequences of tokens (as in [78, 49]) or nodes in graphs [50], such deformation may result in a change of the shape of time series data, which is known to be important for prediction [66]. As a toy example, consider an input time series $\mathbf{x}_i$ where the predictive pattern is driven by feature $\mathbf{x}_i[t_1, k_1]$ is larger than all other features. Suppose $M_{\mathcal{X}}$ is continuous. In that case, it is possible that for a less important feature $\mathbf{x}_i[t_2, k_2]$, $M_{\mathcal{X}}[t_1, k_1] < M_{\mathcal{X}}[t_2, k_2]$ while $(M_{\mathcal{X}}[t_1, k_1] \odot \mathbf{x}_i[t_1, k_1]) > (M_{\mathcal{X}}[t_2, k_2] \odot \mathbf{x}_i[t_2, k_2])$, thereby preserving the predictive pattern. At the same time, the mask indicates that $\mathbf{x}_i[t_2, k_2]$ is more important than $\mathbf{x}_i[t_1, k_1]$. If a surrogate model is trained on $M_{\mathcal{X}} \odot \mathbf{x}_i$, $M_{\mathcal{X}}$ may violate the ordinality expected by an attribution map as defined in Section 3. Discrete masking alleviates this issue by forcing $M_{\mathcal{X}}$ to be binary, removing the possibility of confounds created by continuous masking. Therefore, discrete masking is necessary when learning interpretable masks on continuous time series.

## 4.3 Model behavior consistency

The challenge lies in training $G^E(H^E(\cdot))$ to faithfully represent $F(G(\cdot))$. We approach this by considering the latent spaces of $G$ and $G^E$. If $G$ considers $\mathbf{x}_i$ and $\mathbf{x}_j$ to be similar in $Z$, we expect that a faithful $G^E$ would encode $\mathbf{x}_i^m$ and $\mathbf{x}_j^m$ similarly. However, directly aligning $G$ and $G^E$ is not optimal due to potential differences in the geometry of the explanation embedding space compared to the full input latent space. To address this, we introduce model behavior consistency (MBC). This novel self-supervised objective trains the explainer model to mimic the behavior of the original model without strict alignment between the spaces. Denote the latent space induced by $G$ and $G^E$ as $Z$ and $Z^E$, respectively. The MBC objective is thus defined as:

$$\mathcal{L}_{\text{MBC}}(Z, Z^E) = \frac{1}{N^2} \sum_{\mathbf{z}_i, \mathbf{z}_j \in Z} \sum_{\mathbf{z}_i^E, \mathbf{z}_j^E \in Z^E} (D_Z(\mathbf{z}_i, \mathbf{z}_j) - D_{Z^E}(\mathbf{z}_i^E, \mathbf{z}_j^E))^2, \tag{3}$$

where $D_Z$ and $D_{Z^E}$ are distance functions on the reference model's latent space and the explanation encoder's latent space, respectively, and $N$ is the size of the minibatch, thereby making $N^2$ equal to the number of pairs on which $\mathcal{L}_{\text{MBC}}$ is optimized. This objective encourages distances to be similar across both spaces, encouraging $Z^E$ to retain a similar local topology to $Z$ without performing a direct alignment. This is closely related to cycle-consistency loss, specifically cross-modal cycle-consistency loss as [79]. We use cosine similarity for $D_Z$ and $D_{Z^E}$ throughout experiments in this study, but any distance can be defined on each respective space.

In addition to MBC, we use a label consistency (LC) objective to optimize TIMEX. We train a predictor $F^E$ on $Z^E$ to output logits consistent with those output by $F$. We use a Jensen-Shannon Divergence ($D_{\text{JS}}$) between the logits of both predictors:

$$\mathcal{L}_{\text{LC}}(Z, Z^E) = \sum_{\mathbf{z}_i, \mathbf{z}_j \in Z} \sum_{\mathbf{z}_i^E, \mathbf{z}_j^E \in Z^E} \left( D_{\text{JS}}(F(\mathbf{z}_i) || F(\mathbf{z}_j)) - D_{\text{JS}}(F^E(\mathbf{z}_i^E) || F^E(\mathbf{z}_j^E)) \right)^2 \tag{4}$$

Our total loss function on $Z^E$ can then be defined as a combination of losses: $\mathcal{L}_{Z^E} = \mathcal{L}_{\text{MBC}} + \lambda_{\text{LC}} \mathcal{L}_{\text{LC}}$.

**Consistency learning justification.** MBC offers three critical benefits for explainability. ① MBC enables consistency optimization across two latent spaces $Z$ and $Z^E$ without requiring that both $\mathbf{x}_i$ and $\mathbf{x}_i^m$ be encoded by the same model, allowing the learning of $E$ on a separate model $F^E(G^E(\cdot)) \neq F(G(\cdot))$. This avoids the out-of-distribution problems induced by directly masking inputs to $G$. ② MBC comprehensively represents model behavior for explainer optimization. This is in contrast to perturbation explanations [38, 80, 58] which seek a label-preserving perturbation $P$ on $F(G(\cdot))$ where $F(G(P(\mathbf{x}_i))) \approx F(G(\mathbf{x}_i))$. By using $G(\mathbf{x}_i)$ and $F(G(\mathbf{x}_i))$ to capture the behavior of the reference model, MBC's objective is richer than a simple label-preserving objective. ③ While MBC is stronger than label matching alone, it is more flexible than direct alignment. An alignment objective, which enforces $\mathbf{z}_i \approx \mathbf{z}_i^E$, inhibits $G^E$ from learning important features of explanations not

represented in $Z$. The nuance and novelty of MBC are in learning a latent space that is faithful to model behavior while being flexible enough to encode rich relational structure about explanations that can be exploited to learn additional features such as landmark explanations. Further discussion of the utility of MBC is in Appendix B.

## 4.4 Learning explanation landmarks and training TIMEX models

Leveraging the latent space, TIMEX generates landmark explanations $\mathbf{z}^L \in \mathbb{R}^{d_z}$. Such landmarks are desirable as they allow users to compare similar explanation patterns across samples used by the predictor. Landmarks are learned by a landmark consistency loss, and their optimization is detached from the gradients of the explanations so as not to harm explanation quality. Denote the landmark matrix as $\mathbf{L} \in \mathbb{R}^{n_L \times d_z}$ where $n_L$ corresponds to the number of landmarks (a user-chosen value) and $d_z$ is the dimensionality of $Z^E$. For each sample explanation embedding $\mathbf{z}_i^E$, we use Gumbel-Softmax STE (GS) to stochastically match $\mathbf{z}_i^E$ to the nearest landmark in the embedding space. Denote the vector of similarities to each $\mathbf{z}_i^E$ as $s(\mathbf{z}_i^E, \mathbf{L})$. Then the assimilation $A$ is described as:

$$A(\mathbf{z}_i^E; \mathbf{L}) = \text{GS}(\text{softmax}(s(\text{sg}(\mathbf{z}_i^E), \mathbf{L})))\mathbf{L}, \tag{5}$$

where sg denotes the stop-grad function. The objective for learning landmarks is then $\mathcal{L}_{\text{MBC}}(Z, A(Z^E; \mathbf{L}))$, optimizing the consistency between the assimilated prototypes and the reference model's latent space. Landmarks are initialized as a random sample of explanation embeddings from $Z^E$ and are updated via gradient descent. After learning landmarks, we can measure the quality of each landmark by the number of $\mathbf{z}_i^E$ embeddings closest to it in latent space. We filter out any landmarks not sufficiently close to any samples (described in Appendix B).

**TIMEX training.** The overall loss function for TIMEX has four components: $\mathcal{L} = \mathcal{L}_{\text{MBC}} + \lambda_{\text{LC}}\mathcal{L}_{\text{LC}} + \lambda_E(\mathcal{L}_m + \lambda_{\text{con}}\mathcal{L}_{\text{con}})$, where $\lambda_{\text{LC}}, \lambda_E, \lambda_{\text{con}} \in \mathbb{R}$ are weights for the label consistency loss, total explanation loss, and connective explanation loss, respectively. TIMEX can be optimized end-to-end, requiring little hyperparameter choices from the user. The user must also choose the $r$ parameter for the explanation regularization. Explanation performance is stable across choices of $r$ (as found in [50]), so we set $r = 0.5$ to remain consistent throughout experiments. A lower $r$ value may be provided if the underlying predictive signal is sparse; this hyperparameter is analyzed in Appendix C.3. In total, TIMEX optimizes $H^E$, $G^E$, and $F^E$.

## 5 Experimental setup

**Datasets.** We design four synthetic datasets with known ground-truth explanations: **FreqShapes**, **SeqComb-UV**, **SeqComb-MV**, and **LowVar**. Datasets are designed to capture diverse temporal dynamics in both univariate and multivariate settings. We employ four datasets from real-world time series classification tasks: **ECG** [81] - ECG arrhythmia detection; **PAM** [82] - human activity recognition; **Epilepsy** [83] - EEG seizure detection; and **Boiler** [84] - mechanical fault detection. We define ground-truth explanations for ECG as QRS intervals based on known regions of ECG signals where arrhythmias can be detected. The R, P, and T wave intervals are extracted following [85]. Dataset details are given in Appendix C.1 and C.4.

**Baselines.** We evaluate the method against five explainability baselines. As a general explainer, we use integrated gradients (**IG**) [31]; for recent time series-specific explainers, we use **Dynamask** [58], and **WinIT** [86]; for an explainer that uses contrastive learning, we use **CoRTX** [87]; and for an *in-hoc* explainer which has been demonstrated for time series, we use **SGT + Grad** [17].

**Evaluation.** We consider two approaches. *Ground-truth explanations:* Generated explanations are compared to ground-truth explanations, *i.e.*, known predictive signals in each input time series sample when interpreting a strong predictor, following established setups [6]. We use the area under precision (AUP) and area under recall (AUR) curves to evaluate the quality of explanations [58]. We also use the explanation AUPRC, which combines the results of AUP and AUR. For all metrics, higher values are better. Definitions of metrics are in Appendix C.4. *Feature importance under occlusion:* We occlude the bottom $p$-percentile of features as identified by the explainer and measure the change in prediction AUROC (Sec. 4.2). The most essential features a strong explainer identifies should retain prediction performance under occlusion when $p$ is high. To control for potential misinterpretations, we include a random explainer reference. Our experiments use transformers [88] with time-based positional encoding. Hyperparameters and compute details are given in Appendix C.

| Method | FreqShapes | | | SeqComb-UV | | |
|---|---|---|---|---|---|---|
| | AUPRC | AUP | AUR | AUPRC | AUP | AUR |
| IG | 0.7516±0.0032 | 0.6912±0.0028 | 0.5975±0.0020 | 0.5760±0.0022 | 0.8157±0.0023 | 0.2868±0.0023 |
| Dynamask | 0.2201±0.0013 | 0.2952±0.0037 | 0.5037±0.0015 | 0.4421±0.0016 | 0.8782±0.0039 | 0.1029±0.0007 |
| WinIT | 0.5071±0.0021 | 0.5546±0.0026 | 0.4557±0.0016 | 0.4568±0.0017 | 0.7872±0.0027 | 0.2253±0.0016 |
| CoRTX | 0.6978±0.0156 | 0.4938±0.0004 | 0.3261±0.0012 | 0.5643±0.0024 | 0.8241±0.0025 | 0.1749±0.0007 |
| SGT + Grad | 0.5312±0.0019 | 0.4138±0.0011 | 0.3931±0.0015 | 0.5731±0.0021 | 0.7828±0.0013 | 0.2136±0.0008 |
| TIMEX | **0.8324**±0.0034 | **0.7219**±0.0031 | **0.6381**±0.0022 | **0.7124**±0.0017 | **0.9411**±0.0006 | **0.3380**±0.0014 |

Table 1: Attribution explanation performance on univariate synthetic datasets.

| Method | SeqComb-MV | | | LowVar | | |
|---|---|---|---|---|---|---|
| | AUPRC | AUP | AUR | AUPRC | AUP | AUR |
| IG | 0.3298±0.0015 | 0.7483±0.0027 | 0.2581±0.0028 | **0.8691**±0.0035 | 0.4827±0.0029 | 0.8165±0.0016 |
| Dynamask | 0.3136±0.0019 | 0.5481±0.0053 | 0.1953±0.0025 | 0.1391±0.0012 | 0.1640±0.0028 | 0.2106±0.0018 |
| WinIT | 0.2809±0.0018 | 0.7594±0.0024 | 0.2077±0.0021 | 0.1667±0.0015 | 0.1140±0.0022 | 0.3842±0.0017 |
| CoRTX | 0.3629±0.0021 | 0.5625±0.0006 | 0.3457±0.0017 | 0.4983±0.0014 | 0.3281±0.0027 | 0.4711±0.0013 |
| SGT + Grad | 0.4893±0.0005 | 0.4970±0.0005 | **0.4289**±0.0018 | 0.3449±0.0010 | 0.2133±0.0029 | 0.3528±0.0015 |
| TIMEX | **0.6878**±0.0021 | **0.8326**±0.0008 | 0.3872±0.0015 | 0.8673±0.0033 | **0.5451**±0.0028 | **0.9004**±0.0024 |

Table 2: Attribution explanation performance on multivariate synthetic datasets.

# 6   Results

**R1: Comparison to existing methods on synthetic and real-world datasets.**

**Synthetic datasets.** We compare TIMEX to existing explainers based on how well they can identify essential signals in time series datasets. Tables 1-2 show results for univariate and multivariate datasets, respectively. Across univariate and multivariate settings, TIMEX is the best explainer on 10/12 (3 metrics in 4 datasets) with an average improvement in the explanation AUPRC (10.01%), AUP (6.01%), and AUR (3.35%) over the strongest baselines. Specifically, TIMEX improves ground-truth explanation in terms of AUP by 3.07% on FreqShapes, 6.3% on SeqComb-UV, 8.43% on SeqComb-MV, and 6.24% on LowVar over the strongest baseline on each dataset. In all of these settings, AUR is less critical than AUP since the predictive signals have redundant information. TIMEX achieves high AUR because it is optimized to output smooth masks over time, tending to include larger portions of entire subsequence patterns than sparse portions, which is relevant for human interpretation. To visualize this property, we show TIMEX's explanations in Appendix C.5.

**Real-world datasets: arrhythmia detection.** We demonstrate TIMEX on ECG arrhythmia detection. TIMEX's attribution maps show a state-of-the-art performance for finding relevant QRS intervals driving the arrhythmia diagnosis and outperform the strongest baseline by 5.39% (AUPRC) and 9.83% (AUR) (Table 3). Integrated gradients achieve a slightly higher AUP, whereas state-of-the-art time series explainers perform poorly. Notably, TIMEX's explanations are significantly better in AUR, identifying larger segments of the QRS interval rather than individual timesteps.

**Ablation study on ECG data.** We conduct ablations on the ECG data using TIMEX (Table 3). First, we show that the STE improves performance as opposed to soft attention masking, resulting in an AUPRC performance gain of 9.44%; this validates our claims about the pitfalls of soft masking for time series. Note that this drop in performance becomes more significant when including direct-value masking, as shown in Appendix C.6. Second, we use SimCLR loss to align $Z^E$ to $Z$ as opposed to MBC; SimCLR loss can achieve comparable results in AUPRC and AUR, but the AUP is 13.6% lower than the base TIMEX. Third, we experiment with the usefulness of MBC and LC objectives. MBC alone produces poor explanations with AUPRC at 65.8% lower score than the base model. LC alone does better than MBC alone, but its AUPRC is still 21.5% lower than the base model. MBC and LC, in conjunction, produce high-quality explanations, showing the value in including more intermediate states for optimizing $G^E(H^E(\cdot))$. Extensive ablations are provided in Appendix C.6.

**R2: Occlusion experiments on real-world datasets.**

We evaluate TIMEX explanations by occluding important features from the reference model and observing changes in classification [63, 58, 87]. Given a generated explanation $E(\mathbf{x}_i)$, the bottom $p$-percentile of features are occluded; we expect that if the explainer identifies important features for the model's prediction, then the classification performance to drop significantly when replacing these important features (determined by the explainer) with baseline values. We compare the performance under occlusion to random explanations to counter misinterpretation (Sec. 3.1). We adopt the masking

| Method | ECG AUPRC | ECG AUP | AUR | TIMEX Ablations | ECG AUPRC | ECG AUP | AUR |
|---|---|---|---|---|---|---|---|
| IG | 0.4182±0.0014 | **0.5949**±0.0023 | 0.3204±0.0012 | Full | 0.4721±0.0018 | 0.5663±0.0025 | 0.4457±0.0018 |
| Dynamask | 0.3280±0.0011 | 0.5249±0.0030 | 0.1082±0.0080 | −STE | 0.4014±0.0019 | 0.5570±0.0032 | 0.1564±0.0007 |
| WinIT | 0.3049±0.0011 | 0.4431±0.0026 | 0.3474±0.0011 | +SimCLR | 0.4767±0.0021 | 0.4895±0.0024 | 0.4779±0.0013 |
| CoRTX | 0.3735±0.0008 | 0.4968±0.0021 | 0.3031±0.0009 | Only LC | 0.3704±0.0018 | 0.3296±0.0019 | 0.5084±0.0008 |
| SGT + Grad | 0.3144±0.0010 | 0.4241±0.0024 | 0.2639±0.0013 | Only MBC | 0.1615±0.0006 | 0.1348±0.0006 | 0.5504±0.0011 |
| TIMEX | **0.4721**±0.0018 | 0.5663±0.0025 | **0.4457**±0.0018 | | | | |

Table 3: (*Left*) Benchmarking TIMEX on the ECG dataset. (*Right*) Results of ablation analysis.

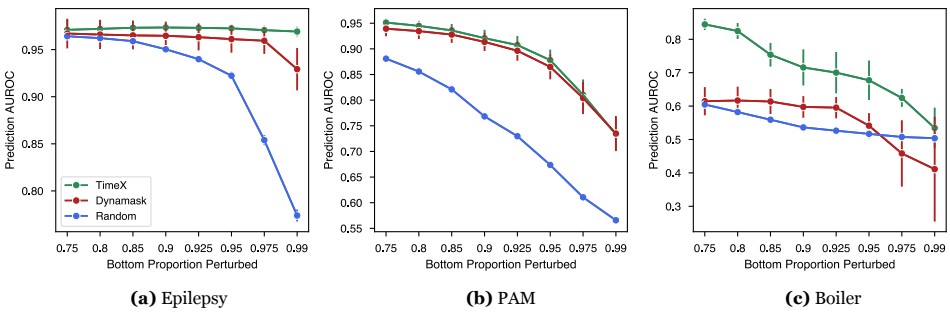

**(a)** Epilepsy    **(b)** PAM    **(c)** Boiler

Figure 3: Occlusion experiments on real-world datasets. Higher values indicate better performance.

procedure described in Sec. 4.2, performing attention masking where applicable and direct-value masking otherwise.

Figure 3 compares TIMEX to Dynamask, a strong time-series explainer. On all datasets, TIMEX's explanations are either at or above the performance of Dynamask, and both methods perform above the random baseline. On the Boiler dataset, we demonstrate an average of 27.8% better classification AUROC across each threshold than Dynamask, with up to 37.4% better AUROC at the 0.75 threshold. This gap in performance between TIMEX and Dynamask is likely because the underlying predictor for Boiler is weaker than that of Epilepsy or PAM, achieving 0.834 AUROC compared to 0.979 for PAM and 0.939 for Epilepsy. We hypothesize that TIMEX outperforms Dynamask because it only considers changes in predicted labels under perturbation while TIMEX optimizes for consistency across both labels and embedding spaces in the surrogate and reference models. TIMEX performs well across both univariate (Epilepsy) and multivariate (PAM and Boiler) datasets.

**R3: Landmark explanation analysis on ECG.**

To demonstrate TIMEX's landmarks, we show how landmarks serve as summaries of diverse patterns in an ECG dataset. Figure 4 visualizes the learned landmarks in the latent space of explanations. We choose four representative landmarks based on the previously described landmark ranking strategy (Sec. 4.4). Every landmark occupies different regions of the latent space, capturing diverse types of explanations generated by the model. We show the three nearest explanations for the top two landmarks regarding the nearest neighbor in the latent space. Explanations ①, ②, and ③ are all similar to each other while distinctly different from ④, ⑤, and ⑥, both in terms of attribution and temporal structure. This visualization shows how landmarks can partition the latent space of explanations into interpretable temporal patterns. We demonstrate the high quality of learned landmark explanations through a quantitative experiment in Appendix C.10.

**Additional experiments demonstrating flexibility of TIMEX.**

In the Appendix, we present several additional experiments to demonstrate the flexibility and superiority of TIMEX. In Appendix C.8, we replicate experiments on two other time series architectures, LSTM and CNN, and show that TIMEX retains state-of-the-art performance. In Appendix C.9, we demonstrate the performance of TIMEX in multiple task settings, including forecasting and irregularly-sampled time series classification. TIMEX can be adapted to these settings with minimal effort and retains excellent explanation performance.

## 7 Conclusion

We develop TIMEX, an interpretable surrogate model for interpreting time series models. By introducing the novel concept of model behavior consistency (*i.e.*, preserving relations in the latent space induced by the pretrained model when compared to relations in the latent space induced by the explainer), we ensure that TIMEX mimics the behavior of a pretrained time series model, aligning

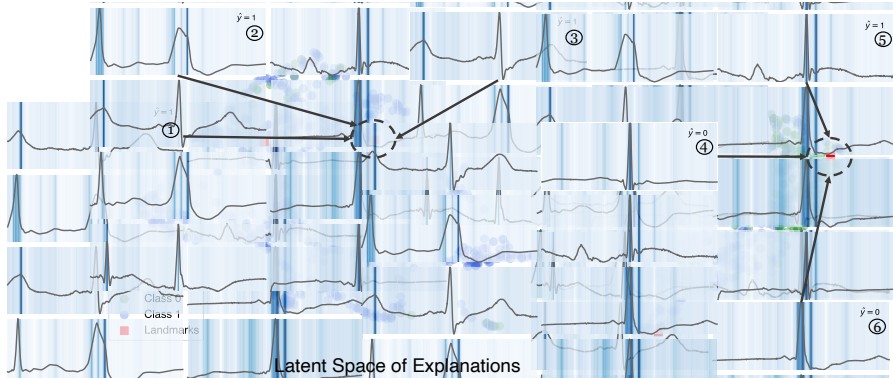

Figure 4: Landmark analysis of TIMEX on the ECG dataset. Shown is a UMAP plot of the latent explanation space along with learned landmark explanations. For two selected landmarks (in red), we show three explanation instances most similar to each landmark.

influential time series signals with interpretable temporal patterns. The generation of attribution maps and utilizing a latent space of explanations distinguish TIMEX from existing methods. Results on synthetic and real-world datasets and case studies involving physiological time series demonstrate the superior performance of TIMEX compared to state-of-the-art interpretability methods. TIMEX's innovative components offer promising potential for training interpretable models that capture the behavior of pretrained time series models.

**Limitations.** While TIMEX is not limited to a specific task as an explainer, our experiments focus on time series classification. TIMEX can explain other downstream tasks, assuming we can access the latent pretrained space, meaning it could be used to examine general pretrained models for time series. Appendix C.9 gives experiments on various setups. However, the lack of such pretrained time series models and datasets with reliable ground-truth explanations restricted our testing in this area. One limitation of our approach is its parameter efficiency due to the separate optimization of the explanation-tuned model. However, we conduct a runtime efficiency test in Appendix C.7 that shows TIMEX has comparable runtimes to baseline explainers. Larger models may require adopting parameter-efficient tuning strategies.

**Societal impacts.** Time series data pervades critical domains including finance, healthcare, energy, and transportation. Enhancing the interpretability of neural networks within these areas has the potential to significantly strengthen decision-making processes and foster greater trust. While explainability plays a crucial role in uncovering systemic biases, thus paving the way for fairer and more inclusive systems, it is vital to approach these systems with caution. The risks of misinterpretations or an over-reliance on automated insights are real and substantial. This underscores the need for a robust framework that prioritizes human-centered evaluations and fosters collaboration between humans and algorithms, complete with feedback loops to continually refine and improve the system. This approach will ensure that the technology serves to augment human capabilities, ultimately leading to more informed and equitable decisions across various sectors of society.

## Acknowledgements

We gratefully acknowledge the support of the Under Secretary of Defense for Research and Engineering under Air Force Contract No. FA8702-15-D-0001 and awards from NIH under No. R01HD108794, Harvard Data Science Initiative, Amazon Faculty Research, Google Research Scholar Program, Bayer Early Excellence in Science, AstraZeneca Research, Roche Alliance with Distinguished Scientists, Pfizer Research, Chan Zuckerberg Initiative, and the Kempner Institute for the Study of Natural and Artificial Intelligence at Harvard University. Any opinions, findings, conclusions, or recommendations expressed in this material are those of the authors and do not necessarily reflect the views of the funders. The authors declare that there are no conflicts of interest.

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

# Appendix A    Further discussion of background

**Straight-through estimators.** Discrete operations, such as thresholding, are often avoided in neural network architectures due to difficulties differentiating discrete functions. To circumvent these issues, [76] introduces the straight-through estimator (STE), which uses a surrogate function during backpropagation to approximate the gradient for a non-differentiable operation. STEs have seen usage in quantized neural networks [S1]. This method shows empirical performance even though there is little theoretical justification behind it [S2].

**Self-supervised learning.** Methods in self-supervised learning (SSL) have become a common pretraining technique for settings in which large, unlabeled datasets are available [S3, S4, S5, S6]. Common approaches for self-supervised learning are contrastive learning, which seeks to learn representations for samples under invariant data augmentations, and metric learning, which aims to generate a latent space in which a distance function captures some pre-defined relations on data [S7]. Consistency learning has emerged as another promising SSL approach; intuitively, this family of methods seeks to learn latent spaces in which similar pairs are expected to be embedded similarly, *i.e.*, preserving some consistent properties. Consistency learning has seen use in aligning videos [S8], enhancing latent geometry for multimodal contrastive learning [79], and pretraining time series models across time and frequency domains [1].

**The use of consistency for explaining machine learning models.** Consistency has been considered in previous XAI literature in two ways: 1) consistency between explanations and 2) consistency as an explainability metric, as explained below.

- **Consistency between explanations**: This notion has been introduced in previous works in explainability literature. Pillai et al. [S9] train a saliency explainer via contrastive learning that preserves consistency across the saliency maps for augmented versions of images. A few other works have explored maintaining consistency of explanations across various perturbations and augmentations, specifically in computer vision [S10, S11]. In one of the only previous works to consider explanation consistency in time series, Watson et al. [S12] train an explainer on an ensemble of classifiers to optimize the consistency of explanations generated by an explainer applied to each classifier. TIMEX does not seek to optimize consistency between explanations but rather a consistency to the predictor model on which it is explaining.

- **Consistency as an explainability metric**: Dasgupta et al. [S13] defines explanation consistency as similar explanations corresponding to similar predictions; this metric is then used as a proxy to faithfulness to evaluate the quality of explainers. However, Dasgupta et al. use consistency to evaluate explainers, not to train and design a new explainer method. TIMEX uses consistency as a learning objective rather than simply a metric.

Our work differs from these previous formulations of explanation consistency. We seek to optimize the consistency not between explanations directly, as mentioned in previous works, but rather between the explainer and the model it is tasked with explaining. MBC attempts to ensure that the behavior of the explainable surrogate matches that of the original model. The definition of consistency in Dasgupta et al. is the closest to our definition of MBC; however, Dasgupta et al. seek not to optimize the consistency of explainers but rrather to evaluate the output of post-hoc explainers. TIMEX directly optimizes the consistency between the surrogate model and the original predictor through the MBC loss, a novel formulation that seeks to increase the faithfulness of explanations generated by TIMEX.

# Appendix B    Further theoretical discussions

## B.1    Differentiable attention masking

As is described in Section 4.1, we use differentiable attention masking [77], which is defined as such:

$$\alpha^m = (\text{softmax}(\frac{\mathbf{Q}\mathbf{K}^T}{\sqrt{d_k}}) \odot \mathbf{M}_\mathcal{X})\mathbf{V}, \tag{6}$$

where $\mathbf{Q}, \mathbf{K}, \mathbf{V}$ represent query, key and values operators, $d_k$ is used as a normalization factor, and $\mathbf{M}_\mathcal{X}$ is a mask with self-attention values. This procedure is fully differentiable, and given that $\mathbf{M}_\mathcal{X}$ is binarized via the STE, it sets all attention values to zero that are to be ignored based on output from $H^E$.

## B.2 Further discussion on the utility of model behavior consistency

The model behavior consistency (MBC) framework in TIMEX is a method to train an interpretable surrogate model $G^E$. In Section 4.3, we discuss the conceptual advances of this approach. Here, we will outline another advantage of the approach—preserving classification performance—and a brief discussion on the broader uses of MBC in other domains and applications.

Training a model with an interpretability bottleneck such as TIMEX is often challenging, as the inherent interpretability mechanism can hinder the performance and expressiveness of the method; this is an advantage of *post-hoc* methods. MBC allows one to preserve the performance of the underlying predictor. TIMEX, a surrogate method, allows one to keep the predictions from a pretrained time series encoder and develop explanations on top of it, which is practical for real-world use when a drop in classification performance is highly undesirable.

MBC is not limited to time series classification tasks. We demonstrate the utility of MBC for time series due to the particularly challenging nature of the data modality and the lack of available time series explainers. However, MBC gives a general framework for learning interpretable surrogate models through learning the $H^E$ and $G^E$ modules. MBC also has the potential to be applied to tasks outside of classification; since MBC is defined on the embedding space, any model with such an embedding space could be matched through a surrogate model as in TIMEX. This opens the possibility of learning on general pretrained models or even more complex tasks such as forecasting (as shown in Appendix C.9). Finally, we see MBC as having potential beyond explainability as well; one could imagine MBC being a way to distill knowledge into smaller models [S14, S15, S16]. We leave these discussions and experiments for future work.

## B.3 Explanation landmark selection strategy

We proceed to describe how landmarks are selected for final interpretation. As described in Section 4.4, landmarks are initialized with the embeddings given by $G$ for a random number of training samples. Practically, we stratify this selection across classes in the training set. Landmarks are then updated during the learning procedure. After learning landmarks, not every landmark will be helpful as an explanation; thus, we perform a filtration procedure. Intuitively, this filtration consists of detecting landmarks for which the landmark is the nearest landmark neighbor for many samples. This procedure is described in Algorithm 1.

---

**Algorithm 1:** Landmark filtration

---

**Input**: Landmark matrix $\mathbf{L} \in \mathbb{R}^{n_L \times d_z}$; training explanation embeddings $\{\mathbf{z}_1^E, ..., \mathbf{z}_N^E\}$ for $\mathbf{z}_i^E \in \mathbb{R}^{d_z}$; threshold number of neighbors $n_\epsilon \in \mathbb{N}$
$N_l \leftarrow \{\}$
**for** $i \leftarrow 1$ ***to*** $N$ **do**
    Compute similarity to all landmarks $S_i^l = \mathrm{sim}(\mathbf{z}_i^E, \mathbf{L})$
    $j_{\max} \leftarrow \mathrm{argmax}_j\, S_i^l[j]$ (Gets nearest landmark for sample explanation $i$)
    Append $j_{\max}$ to $N_l$
**end**
$F_l \leftarrow$ Frequency of occurrence of each unique element in $N_l$
$\mathbf{L}_{\mathrm{filter}} \leftarrow$ every landmark in $\mathbf{L}$ s.t. $F_l \geq n_\epsilon$
Return $\mathbf{L}_{\mathrm{filter}}$

---

# Appendix C   Additional experiments and experimental details

## C.1 Description of datasets

We conduct experiments using both synthetic and real-world datasets. This section describes each synthetic and real-world dataset, including how ground-truth explanations are generated when applicable.

### C.1.1 Synthetic datasets

We employ synthetic datasets with known ground-truth explanations to study the capability to identify the underlying predictive signal. We follow standard practices for designing synthetic datasets, including tasks that are predictive and not susceptible to shortcut learning [S17] induced by logical shortcuts. These principles are defined in [S18] concerning graphs, but we extend these to synthetic datasets for time series. Each time series is initialized with a non-autoregressive moving average (NARMA) noise base, and then the described patterns are inserted. We will briefly describe the construction of each time series dataset in this section, and the codebase contains full details at `https://github.com/mims-harvard/TimeX`. We designed four synthetic datasets to test different time series dynamics:

**FreqShapes.** Predictive signal is determined by the frequency of occurrence of an anomaly signal. To construct the dataset, take two upward and downward spike shapes and two frequencies, 10 and 17 time steps. There are four classes, each with a different combination of the attributes: class 0 has a downward spike occurring every 10-time steps, class 1 has an upward spike occurring every 10-time steps, class 2 has a downward spike occurring every 17-time steps, and class 3 has an upward spike occurring every 17-time steps. Ground-truth explanations are the locations of the upward and downward spikes.

**SeqComb-UV.** Predictive signal is defined by the presence of two shapes of subsequences: increasing (I) and decreasing (D) trends. First, two subsequence regions are chosen within the time series so neither subsequence overlaps; each subsequence is 10-20 time steps long. Then, a pattern is inserted based on the class identity; the increasing or decreasing trend is created with a sinusoidal noise with a randomly-chosen wavelength. Class 0 is null, following a strategy in [S18] that recommends using null classes for simple logical identification tasks in synthetic datasets. Class 1 is I, I; class 2 is D, D; and class 3 is I, D. Thus, the model is tasked with identifying both subsequences to classify each sample. Ground-truth explanations are the I and D sequences determining class labels.

**SeqComb-MV.** This dataset is a multivariate version of **SeqComb-UV**. The construction and class structure are equivalent, but the I and D subsequences are distributed across different sensors in the input. Upon constructing the samples, the subsequences are chosen to be on random sensors throughout the input. Ground-truth explanations are given as the predictive subsequences on their respective sensors, *i.e.*, the explainer is required to identify the time points at which the causal signal occurs and the sensors upon which they occur.

**LowVar.** Predictive signal is defined by regions of low variance over time that occur in a multivariate time series sample. Similar to **SeqComb** datasets, we choose a random subsequence in the input and, in that subsequence, replace the NARMA background sequence with Gaussian noise at a low variance. The subsequence is further discriminated by the mean of the Gaussian noise and the sensor on which the low variance sequence occurs. For class 0, the subsequence is at mean -1.5 on sensor 0; for class 1, the subsequence is at mean 1.5 on sensor 0; for class 2, the subsequence is at mean -1.5 on sensor 1; for class 3, the subsequence is at mean 1.5 on sensor 1. This task is distinctly different from other synthetic datasets, requiring recognition of a subsequence that is not anomalous from the rest of the sequence. This presents a more challenging explanation task; a simple change-point detection algorithm could not determine the explanation for this dataset.

We create 5,000 training samples, 1,000 testing samples, and 100 validation samples for each dataset. A summary of the dimensions of each dataset can be found in Table 4.

Table 4: Synthetic Dataset Description

| Dataset | # of Samples | Length | Dimension | Classes |
|---------|-------------|--------|-----------|---------|
| FreqShapes | 6,100 | 50 | 1 | 4 |
| SeqComb-UV | 6,100 | 200 | 1 | 4 |
| SeqComb-MV | 6,100 | 200 | 4 | 4 |
| LowVarDetect | 6,100 | 200 | 2 | 4 |

### C.1.2 Real-world datasets

We employ four datasets from real-world time series classification tasks: **PAM** [82] - human activity recognition; **ECG** [81] - ECG arrhythmia detection; **Epilepsy** [83] - EEG seizure detection; and **Boiler** [84] - automatic fault detection.

**PAM [82].** It measures the daily living activities of 9 subjects with three inertial measurement units. We excluded the ninth subject due to the short length of sensor readouts. We segment the continuous signals into samples with a time window of 600 and the overlapping rate of 50%. PAM initially has 18 activities of daily life. We exclude the ones associated with fewer than 500 samples, leaving us with eight activities. After modification, the PAM dataset contains 5,333 segments (samples) of sensory signals. Each sample is measured by 17 sensors and contains 600 continuous observations with a sampling frequency of 100 Hz. PAM is labeled into eight classes, where each class represents an activity of daily living. PAM does not include static attributes, and the samples are approximately balanced across all eight classes.

**MIT-BIH (ECG) [81].** The MIT-BIH dataset has ECG recordings from 47 subjects recorded at the sampling rate of 360Hz. The raw dataset was then window-sliced into 92511 samples of 360 timestamps each. Two cardiologists have labeled each beat independently. Of the available annotations, we choose to use three for classification: normal reading (N), left bundle branch block beat (L), and right bundle branch block beat (R). We choose these because L and R diagnoses are known to rely on the QRS interval [S19, S20], which will then become our ground-truth explanation (see Section C.4). The Arrhythmia classification problem involves classifying each fragment of ECG recordings into different beat categories.

**Epilepsy [83].** The dataset contains single-channel EEG measurements from 500 subjects. For every subject, the brain activity was recorded for 23.6 seconds. The dataset was then divided and shuffled (to mitigate sample-subject association) into 11,500 samples of 1 second each, sampled at 178 Hz. The raw dataset features five classification labels corresponding to different states of subjects or measurement locations — eyes open, eyes closed, EEG measured in the healthy brain region, EEG measured in the tumor region, and whether the subject has a seizure episode. To emphasize the distinction between positive and negative samples, we merge the first four classes into one, and each time series sample has a binary label indicating whether an individual is experiencing a seizure. There are 11,500 EEG samples in total.

**Boiler [84].** This dataset consists of simulations of hot water heating boilers undergoing different mechanical faults. Various mechanical sensors are recorded over time to derive a time series dataset. The learning task is to detect the mechanical fault of the blowdown valve of each boiler. The dataset is particularly challenging because it includes a large dimension-to-length ratio, unlike the other datasets, which contain many more time steps than sensors (Table 5).

Table 5: Real-World Dataset Description

| Dataset | # of Samples | Length | Dimension | Classes | Task |
|---------|--------------|--------|-----------|---------|------|
| PAM | 5,333 | 600 | 17 | 8 | Action recognition |
| MIT-BIH | 92,511 | 360 | 1 | 5 | ECG classification |
| Epilepsy | 11,500 | 178 | 1 | 2 | EEG classification |
| Boiler | 160,719 | 36 | 20 | 2 | Mechanical fault detection |

### C.2 Descriptions of baseline methods

We now describe each baseline method in further detail.

**IG [31].** Integrated gradients is a classical attribution method that utilizes the gradients of the model to form an explanation. The method compares the gradients to a baseline value and performs Riemannian integration to derive the explanation. Integrated gradients is a popular data-type agnostic interpretability method [S21], but it has no inductive biases specific for time series. We use the Captum [S22] implementation of this method, including default hyperparameters such as the baseline value.

**Dynamask [58].** This explainer is built specifically for time series and uses a perturbation-based procedure to generate explanations. The method performs iterative occlusion of various input portions, learning a mask that deforms the input time series towards a carefully-determined baseline value.

This method is different from TIMEX in a few key ways. First, it performs continuous masking; TIMEX performs discrete masking through STEs. Second, it measures perturbation impact on the original model $F(G(\cdot))$; TIMEX trains a surrogate model $G^E$ to learn the explanations and measure the impact of masking the input. Third, Dynamask learns the explanations iteratively for each sample; TIMEX trains the surrogate, which can then output explanations in one forward pass of $H^E$.

**WinIT [86].** This explainer is a feature removal explainer, similar to Dynamask. WinIT measures the impact of removing features from a time series on the final prediction value. It eliminates the impact of specific time intervals and learns feature dependencies across time steps. WinIT uses a generative model to perform in-distribution replacement of masked-out features. WinIT improves on a previous time series explainer, FIT [63], which is a popular baseline in time series explainability literature but is excluded in our work because WinIT is more recent and improves on FIT both conceptually and empirically.

**CoRTX [87].** Contrastive real-time explainer (CoRTX) is an explainer method that utilizes contrastive learning to approximate SHAP [3] values. This method is developed for computer vision, but we implement a custom version that works with time series encoders and explanation generators. We include this method because it uses self-supervised learning to learn explanations. TIMEX also uses a self-supervised objective to learn explanations, but our method differs from CoRTX in several ways. First, CoRTX performs augmentation-based contrastive learning while we use MBC, which avoids the definition of negatives or the careful choice of augmentations specific to the data modality. Second, CoRTX fundamentally attempts to approximate SHAP values via a small number of SHAP explanations. In contrast, TIMEX includes a masking system that can produce masks without fine-tuning a model on a set of explanations derived from an external method. CorRTX parallels ours in using self-supervised learning but is fundamentally different from TIMEX.

**SGT + Grad [17].** Saliency-guided training (SGT), an *in-hoc* explainer, is based on a modification to the training procedure. During training, features with low gradients are masked to steer the model to focus on more important regions for the prediction. The method is not an explainer alone but requires another *post-hoc* explainer to derive explanations. In our experiments, we consider saliency explanations, which the SGT authors recommend. The authors found that this method can improve performance on time series data. For this reason, we include it as one of our baselines to demonstrate the effectiveness of TIMEX against modern *in-hoc* explainers.

### C.3 Hyperparameter selection

We list hyperparameters for each experiment performed in this work. For the ground-truth attribution experiments (Section 6, results **R1**), the hyperparameters are listed in Table 6. The hyperparameters used for the occlusion experiment (Section 6, results **R2**) with real-world datasets are in Table 7. We also list the architecture hyperparameters for the predictors trained on each dataset in Tables 8-9.

A few abbreviations are used for hyperparameters that are not mentioned in the main text. "Weight decay" refers to an L1 regularization on the model weights; the value for weight decay is equivalent to the weight on that term in the loss function compared to the rest of the loss terms (Section 4.4). "Scheduler?" refers to using a learning rate scheduler that decreases the learning rate by a factor of 10 if a plateau occurs. We use a scheduler that delays decreasing learning rates until after 20 epochs; not every experiment utilizes the scheduler as it is based on which choice yields lower validation loss upon convergence. "Distance norm." refers to a normalization of the distances in $\mathcal{L}_{\text{MBC}}$; the loss is divided by the variance of the distances on the $Z$ embedding space. $\tau$ is the temperature parameter used for the Gumbel-Softmax reparameterization [73], Section 4.1. $d_h$ refers to the dimensionality of hidden layers in the transformer predictor. Finally, "Norm. embedding" refers to an architecture choice that normalizes $Z$ when training the predictor; this is used to prevent a poor latent space when a collapse is observed via poor latent space geometry.

A few other notes on implementation and design of TIMEX: The architecture of $H^E$ uses the same size of $G^E$ and encoder for $H^E$ as for the predictor on each task. The number of transformer decoder layers is fixed at 2. Please reference the codebase for more details on these hyperparameters and implementations https://github.com/mims-harvard/TimeX.

**Choosing the $r$ parameter.** One of the most significant parameters in training TIMEX is $r$, the parameter which controls sparsity of the learned masks. We conduct an experiment where we vary the parameters and measure explanation quality. We use the SeqComb-UV dataset and hold all hyperparameters constant while varying the parameter. The result is visualized in Figure 5. Low

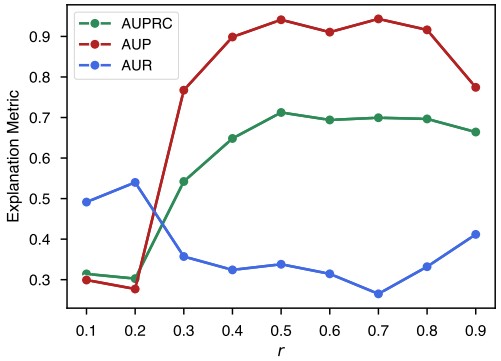

Figure 5: Experiment on SeqComb-UV dataset varying the $r$ parameter.

| Parameter | FreqShape | SeqComb-UV | SeqComb-MV | LowVarDetect | ECG |
|---|---|---|---|---|---|
| Learning rate | 0.001 | 0.001 | 0.001 | 0.003 | 0.0005 |
| Batch size | 64 | 64 | 64 | 64 | 16 |
| Weight decay | 0.001 | 0.001 | 0.001 | 0.0001 | 0.0001 |
| Scheduler? | Yes | Yes | No | No | No |
| Epochs | 50 | 50 | 100 | 100 | 5 |
| $r$ | 0.5 | 0.5 | 0.5 | 0.5 | 0.5 |
| Distance norm. | No | No | No | Yes | No |
| $\lambda_{LC}$ | 1.0 | 1.0 | 1.0 | 1.0 | 1.0 |
| $\lambda_E$ | 2.0 | 2.0 | 2.0 | 2.0 | 2.0 |
| $\lambda_{con}$ | 2.0 | 2.0 | 2.0 | 2.0 | 2.0 |
| $\tau$ | 1.0 | 1.0 | 1.0 | 1.0 | 1.0 |
| $n_L$ | 50 | 50 | 50 | 50 | 50 |

Table 6: Training parameters for TIMEX across all ground-truth attribution experiments.

values lead to a drop in explainer performance with respect to AUPRC and AUP. Importantly, for values above 0.4, the explainer performance is stable, suggesting that is robust to choice of value. Thus, we recommend choosing a value near 0.5 for experiments, as we did throughout our experiments. If the pattern is hypothesized to be sparse, one can set the value lower, and vice versa if the pattern is hypothesized to be dense.

## C.4   Evaluation details

Following [58], we use AUP and AUR to evaluate the goodness of identification of salient attributes as a binary classification task, which is defined in C.1:

**Definition C.1** (AUP,AUR [58]). Let $\mathbf{Q}$ be a matrix in $\{0,1\}^{T \times d_X}$ whose elements indicate the true saliency of the inputs contained in $\mathbf{x} \in \mathbb{R}^{T \times d_X}$. By definition, $Q_{t,i} = 1$ if the feature $x_{t,i}$ is salient

| Parameter | Epilepsy | PAM | Boiler |
|---|---|---|---|
| Learning rate | 0.0001 | 0.002 | 0.0001 |
| Batch size | 32 | 32 | 32 |
| Weight decay | 0.001 | 0.001 | 0.001 |
| Scheduler? | Yes | No | Yes |
| Epochs | 50 | 100 | 50 |
| $r$ | 0.5 | 0.1 | 0.5 |
| Distance norm. | No | Yes | No |
| $\lambda_{LC}$ | 1.0 | 1.0 | 1.0 |
| $\lambda_E$ | 2.0 | 2.0 | 2.0 |
| $\lambda_{con}$ | 2.0 | 0.0 | 2.0 |
| $\tau$ | 1.0 | 1.0 | 1.0 |
| $n_L$ | 50 | 50 | 50 |

Table 7: Training parameters for TIMEX across all real-world datasets used for the occlusion experiments.

| Parameter | FreqShape | SeqComb-UV | SeqComb-MV | LowVarDetect | ECG |
|---|---|---|---|---|---|
| Num. layers | 1 | 2 | 2 | 1 | 1 |
| $d_h$ | 16 | 64 | 128 | 32 | 64 |
| Dropout | 0.1 | 0.25 | 0.25 | 0.25 | 0.1 |
| Norm. embedding | No | No | No | Yes | Yes |
| Learning rate | 0.001 | 0.001 | 5e-4 | 0.001 | 2e-3 |
| Weight decay | 0.1 | 0.01 | 0.001 | 0.01 | 0.001 |
| Epochs | 100 | 200 | 1000 | 120 | 500 |

Table 8: Training parameters for transformer predictors across all ground-truth attribution experiment datasets.

| Param. | Epilepsy | PAM | Boiler |
|---|---|---|---|
| Num. layers | 1 | 1 | 1 |
| $d_h$ | 16 | 72 | 32 |
| Dropout | 0.1 | 0.25 | 0.25 |
| Norm. embedding | No | No | Yes |
| Learning rate | 0.0001 | 0.001 | 0.001 |
| Weight decay | 0.001 | 0.01 | 0.001 |
| Epochs | 300 | 100 | 500 |

Table 9: Training parameters for TIMEX across all real-world datasets used for the occlusion experiments.

and 0 otherwise. Let $M$ be a mask in $\{0,1\}^{T \times d_X}$ obtained with a saliency method. Let $\tau \in (0,1)$ be the detection threshold for $M_{t,i}$ to indicate that the feature $\mathbf{x}_{t,i}$ is salient. This allows to convert the mask into an estimator $\hat{Q}_{t,i}(\tau)$ via:

$$\hat{Q}_{t,i}(\tau) = \left\{ \begin{array}{ll} 1 & \text{if } M_{t,i} \geq \tau \\ 0 & \text{else.} \end{array} \right.$$

By considering the sets of truly salient indexes and the set of indexes selected by the saliency method:

$$A = \{(t,i) \in [1:T] \times [1:d_X] \mid q_{t,i} = 1\}$$
$$\hat{A}(\tau) = \{(t,i) \in [1:T] \times [1:d_X] \mid \hat{q}_{t,i}(\tau) = 1\}.$$

the precision and recall curves that map each threshold to a precision and recall score:

$$\text{P} : (0,1) \longrightarrow [0,1] : \tau \longmapsto \frac{|A \cap \hat{A}(\tau)|}{|\hat{A}(\tau)|}$$

$$\text{R} : (0,1) \longrightarrow [0,1] : \tau \longmapsto \frac{|A \cap \hat{A}(\tau)|}{|A|}.$$

The AUP and AUR scores are the area under these curves:

$$\text{AUP} = \int_0^1 \text{P}(\tau)d\tau$$

$$\text{AUR} = \int_0^1 \text{R}(\tau)d\tau.$$

**Groud-truth explanations for ECG datasets.** We extract ground-truth explanations via a QRS detection strategy following [85] because an initial set of beat labels was produced by a simple slope-sensitive QRS detector and were then given to two cardiologists, who worked on them independently. The cardiologists added additional beat labels where the detector missed beats, deleted false detections as necessary, and changed the labels for all abnormal beats. We employ Neurokit [1] to extract QRS complexes and also take care to ensure that the QRS is the proper explanation for each class. We consider two types of arrhythmias: left bundle branch block beat and right bundle branch block beat, to categorize our "abnormal" class. We perform the ground-truth evaluation on only the abnormal

---
[1] https://github.com/neuropsychology/NeuroKit

| Method | IoU | AUPRC |
|--------|-----|-------|
| IG | 0.3750±0.0022 | 0.5760±0.0022 |
| Dynamask | 0.2958±0.0014 | 0.4421±0.0016 |
| TIMEX | **0.5214**±0.0019 | **0.7124**±0.0017 |

Table 10: IoU metric compared to AUPRC on the SeqComb-UV dataset.

| Method | No STE | | | STE | | |
|--------|--------|-----|-----|-----|-----|-----|
| | AUPRC | AUP | AUR | AUPRC | AUP | AUR |
| FreqShapes | 0.6695±0.0038 | 0.6398±0.0038 | 0.5454±0.0026 | **0.8324**±0.0034 | **0.7219**±0.0031 | **0.6381**±0.0022 |
| SeqComb-MV | 0.5694±0.0023 | **0.8723**±0.0006 | 0.3229±0.0017 | **0.6878**±0.0021 | 0.8326±0.0008 | **0.3872**±0.0015 |
| ECG | 0.4014±0.0019 | 0.5570±0.0032 | 0.1564±0.0007 | **0.4721**±0.0018 | **0.5663**±0.0025 | **0.4457**±0.0018 |

Table 11: **Ablation 1:** Ablation using the STE vs. no STE. "No STE" is equivalent to continuous masking, as discussed in Section 4.2.

class, as the normal class signifies negative information, which may be harder to pinpoint based on model logic.

**Statistical analysis.** We evaluate each experiment on a 5-fold cross-validation of each dataset. We then report average performance and standard error across all folds of evaluation for each experiment, which results in the error bars seen in all tables throughout this work.

**Evaluation with IoU metric.** To further demonstrate the superior performance of TIMEX and the soundness of our evaluation setup, we additionally use an Intersection over Union (IoU) metric to measure the quality of explanations produced by baseline explainers compared to that of TIMEX when compared to ground-truth explanations. We calculate the IoU score on the SeqComb-UV dataset, as shown in Table 10. For comparison to our metrics, we include the AUPRC results for the same methods. The IoU metric is highly correlated with the AUPRC metric, with each metric resulting in the same ranking of methods and TIMEX achieving the highest metric.

## C.5 Visualization of explanations

Figure 6 shows an example of TIMEX explainer versus IG and Dynamask. Shown is the **SeqComb-UV** dataset, which has increasing and decreasing subsequences that determine the class label. Each explainer identifies the regions driving the prediction. IG identifies very sparse portions of the predictive region, choosing only one point out of the sequences for the explanation; this is not reasonable when scaling to large and noisier datasets where the signal might not be as clear. Dynamask seems to miss some important subsequences, identifying one or two subsequences. In contrast, TIMEX identifies a larger portion of the important subsequences due to the connection loss in Equation 2. This property becomes crucial when scaling to time series datasets with more noise as it becomes more difficult to intuitively deduce the causal signal through visual inspection.

## C.6 Further ablation experiments

We present a more in-depth study of ablations on TIMEX on three datasets: FreqShapes (univariate), SeqComb-MV (multivariate), and ECG (real-world). This is an extension to the ablations on the ECG dataset in Section 6, **R1** in Table 3.

**Ablation 1: No STE.** We now conduct an experiment examining the effectiveness of using the STE for training TIMEX. Table 11 shows the results of this ablation experiment. Using the STE provides over a 17% increase in AUPRC for attribution identification for every dataset. Furthermore, AUR is better when using an STE for every dataset, but the AUP is better for SeqComb-MV without the STE than with the STE. Using the STE also shows benefits in both the univariate (FreqShapes, ECG) and multivariate (SeqComb-MV) settings. In conclusion, the STE provides a noticeable benefit over a continuous masking approach, giving empirical evidence for the claims made in Section 4.2.

**Ablation 2: SimCLR vs. MBC.** We now test a classical SimCLR [S5] contrastive learning loss against our proposed model behavior consistency (MBC). The SimCLR objective is designed to decrease the distance between explanation embeddings and embeddings in the reference model's latent space. We *do not* perform data augmentations as in the original SimCLR work. The SimCLR loss that we use is given as:

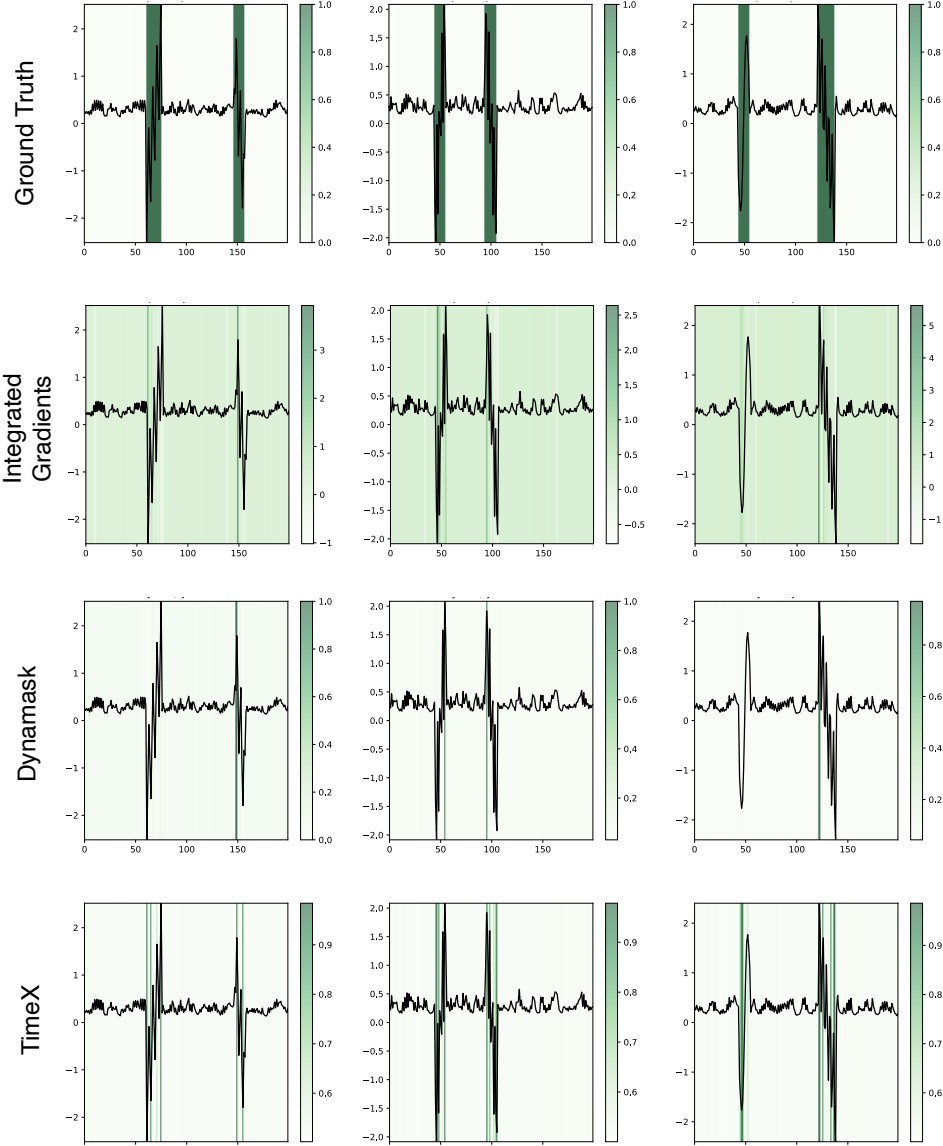

Figure 6: Visualization of explanations on **SeqComb-UV** dataset. Each column corresponds to a unique sample. All are of Class 3, which consists of one increasing subsequence and one decreasing subsequence. The methods that generate each figure are shown for each of the rows, while ground truth explanations are provided in the top row.

$$\mathcal{L}_{\text{SimCLR}}(Z, Z^E) = \frac{1}{N} \sum_{\mathbf{z}_i \in Z, \mathbf{z}_i^E \in Z^E} - \log \frac{\exp(D(\mathbf{z}_i, \mathbf{z}_i^E))}{\sum_{j \neq i} \exp(D(\mathbf{z}_j, \mathbf{z}_i^E))} \tag{7}$$

For each SimCLR trial, we fixed the number of sampled negatives at 32 and kept all other parameters equal. In addition, an early stopping strategy was performed where the stopping value was based on cosine similarity between explanation embeddings and reference sample embeddings (higher similarity is better).

SimCLR loss provides a valuable objective for training TIMEX relative to baseline explainers, but MBC optimization produces more robust explanations. SimCLR delivers a slightly better AUPRC for ECG, but its AUPRC values are below that of MBC for FreqShapes and SeqComb-MV. SimCLR loss yields explanations with consistently lower AUP; AUP is closest for SeqComb-MV with only

| Method | SimCLR | | | MBC | | |
|---|---|---|---|---|---|---|
| | AUPRC | AUP | AUR | AUPRC | AUP | AUR |
| FreqShapes | 0.7014±0.0046 | 0.5991±0.5915 | 0.5915±0.0027 | **0.8324**±0.0034 | **0.7219**±0.0031 | **0.6381**±0.0022 |
| SeqComb-MV | 0.6645±0.0019 | 0.8148±0.0009 | 0.3777±0.0017 | **0.6878**±0.0021 | **0.8326**±0.0008 | **0.3872**±0.0015 |
| ECG | **0.4767**±0.0021 | 0.4895±0.0024 | **0.4779**±0.0013 | 0.4721±0.0018 | **0.5663**±0.0025 | 0.4457±0.0018 |

Table 12: **Ablation 2:** Ablation considering SimCLR objective for training TIMEX versus an MBC objective as outlined in the main text.

| Dataset | Ablation | AUPRC | AUP | AUR |
|---|---|---|---|---|
| FreqShapes | MBC only | 0.2316±0.0020 | 0.1533±0.0015 | 0.4763±0.0022 |
| | LC only | 0.2629±0.0022 | 0.1850±0.0016 | 0.5893±0.0018 |
| | MBC + LC | **0.8324**±0.0034 | **0.7219**±0.0031 | **0.6381**±0.0022 |
| SeqComb-MV | MBC only | 0.0761±0.0008 | 0.0576±0.0006 | 0.4996±0.0019 |
| | LC only | 0.0788±0.0009 | 0.0570±0.0006 | **0.5294**±0.0034 |
| | MBC + LC | **0.6878**±0.0021 | **0.8326**±0.0008 | 0.3872±0.0015 |
| ECG | MBC only | 0.1615±0.0006 | 0.1348±0.0006 | **0.5504**±0.0011 |
| | LC only | 0.3704±0.0018 | 0.3296±0.0019 | 0.5084±0.0008 |
| | MBC + LC | **0.4721**±0.0018 | **0.5663**±0.0025 | 0.4457±0.0018 |

Table 13: **Ablation 3:** Effects of model behavior consistency (MBC) and label consistency (LC) losses on explanation performance.

a 3.4% drop from MBC, but it is at a 17.0% decline for FreqShapes and a 13.6% drop for ECG. It is important to note that in addition to increased performance, MBC loss is more computationally efficient than SimCLR loss, avoiding inference on negative samples.

**Ablation 3: Effect of MBC and LC losses.** We now examine the effectiveness of using both model behavior consistency (Eq. 3) (MBC) and label consistency (Eq. 4) (LC) losses. Table 13 shows that using LC and MBC in combination is always better than using either alone. In isolation, LC performs better than MBC, which is expected given its (obviously) higher correlation with the classification predictions than MBC, which relies on an earlier layer's embedding space. Using both losses results in a powerful explainer that achieves over 27.5% higher AUPRC than MBC or LC alone. MBC and LC work together to capture rich information about the model's behavior, allowing TIMEX to be a state-of-the-art explainer.

To justify these results, we recall an argument presented in Section 4.3, where we justify MBC. We remark that perturbation-based methods have a similar idea to TIMEX: find some sparse perturbation to the input that can preserve the output state of the model. This is often done by observing the predicted label from the model under an applied mask or perturbation, e.g., in one of our baselines, Dynamask. A perturbation that preserves the output state is said to be "faithful" to the model because it is assumed that the model is invariant to the perturbation. In a sense, MBC generalizes this idea to latent spaces, ensuring that invariances are preserved in the latent space of the model as well as the prediction space.

Beyond the introduction of MBC alone, another core contribution of our work focuses on optimizing faithfulness to predictor models on multiple levels. We use multiple hidden or output states of the model, e.g., a latent and logit space, on which the explainable surrogate should match the reference predictor. The hypothesis behind this innovation is that model behavior (the exact objective we are trying to explain) cannot be fully captured by one state, e.g., a predicted label, but rather by multiple states throughout the model. A similar assumption is made in knowledge distillation, where methods often optimize student models to match the teacher in various network layers. Therefore, MBC and LA together enforce adherence to model behavior on two fronts: the latent space and prediction space, respectively. This explains the observed behavior: MBC and LA perform poorly alone, but together, these two losses provide a powerful objective to train the model.

### C.7 Implementation and computing resources

**Implementation.** We implemented all methods in this study using Python 3.8+ and PyTorch 2.0. In our experiments, we employed the Vanilla Transformer [88] as the classification model for all methods. We verified that the classification models achieved satisfactory performance on the testing set to ensure that the models are strong predictors, which was previously pointed out by Faber et al.

[S18] as necessary for explainability evaluation. Complete classification results are in Table 15. We followed the hyperparameters recommended by the respective authors for all baseline methods.

**Computational resources.** For computational resources, we use a GPU cluster with various GPUs, ranging from 32GB Tesla V100s GPU to 48GB RTX8000 GPU. TIMEX and all models were only trained on a single GPU at any given time. The average experiment runtime in this work was around 5 minutes per fold, with ECG taking the longest at approximately 13 minutes per fold when training TIMEX to convergence.

|  | PAM | | Epilepsy | |
|  | Training | Inference | Training | Inference |
| --- | --- | --- | --- | --- |
| Dynamask | N/A | 102 | N/A | 358 |
| WinIT | 1810 | 5730 | 93.3 | 84.7 |
| TIMEX | 580 | 1.04 | 247 | 1.07 |

Table 14: Runtime results for Dynamask, WinIT, and TIMEX on PAM and Epilepsy datasets. Time is shown in seconds. Note that Dynamask requires no training, thus a "N/A" designation.

**Runtime experiment.** We conducted a runtime experiment to understand how TIMEX compares to baseline explainers. Table X shows the training and inference time in seconds of TIMEX versus two state-of-the-art time series-specific baselines, Dynamask and WinIT. We chose two real-world time series datasets, PAM and Epilepsy, which are of varying sizes. PAM contains 4266 training samples and 534 testing samples, each of 600 time steps in length. Epilepsy contains 8280 training samples and 2300 testing samples, each of 178 time steps in length. Table 14 shows the time in seconds needed to train each explainer and to perform inference on the testing set.

TIMEX is the most efficient model at inference time for both datasets. This result is expected, as Dynamask and WinIT both require iterative procedures for each sample at inference time, while TIMEX requires only a forward pass at inference. Combining training and inference time, TIMEX is the second-fastest on both datasets. However, WinIT and Dynamask times vary greatly between each dataset, with Dynamask being the fastest on PAM and WinIT being the fastest on Epilepsy. WinIT scales poorly to samples with many time steps, while Dynamask scales poorly to large testing sets. TIMEX strikes a compromise between these extremes, scaling better than Dynamask to larger testing sets while scaling better than WinIT to longer time series.

## C.8 Flexible use of TIMEX with different time series architectures

We now study the ability of TIMEX to work with different underlying time series architectures. This means that of the original architecture, $G$ and $G^E$ are now an alternative architecture, while $H^E$ remains as described in Section 4.1. Since experiments in the main text are based on transformer architectures, we now use a convolutional neural network (CNN) and long-short term memory (LSTM) network as the underlying predictors with the following hyperparameters:

- LSTM: 3 layer bidirectional LSTM + MLP on mean of last hidden states
- CNN: 3 layer CNN + MLP on meanpool

Tables 18 19 show the results of TIMEX against strong baselines with a CNN predictor. TIMEX retains the state-of-the-art prediction observed for the transformer-based architecture, achieving

| Dataset | F1 | AUPRC | AUROC |
| --- | --- | --- | --- |
| FreqShapes | 0.9716±0.0034 | 0.9940±0.0008 | 0.9980±0.0003 |
| SeqComb-UV | 0.9415±0.0052 | 0.9798±0.0028 | 0.9921±0.0011 |
| SeqComb-MV | 0.9765±0.0024 | 0.9971±0.0005 | 0.9990±0.0001 |
| LowVar | 0.9748±0.0056 | 0.9967±0.0013 | 0.9988±0.0005 |
| Boiler | 0.8345±0.0089 | 0.8344±0.0071 | 0.8865±0.0159 |
| ECG | 0.9154±0.0134 | 0.9341±0.0169 | 0.9587±0.0111 |
| Epilepsy | 0.9201±0.0079 | 0.9246±0.0130 | 0.9391±0.0157 |
| PAM | 0.8845±0.0051 | 0.9251±0.0029 | 0.9786±0.0009 |

Table 15: Classification (*i.e.*, predictive) performance achieved by transformer time series models on datasets used in this study. These models are considered time series predictors throughout experiments in this study.

| Method | FreqShapes | | | SeqComb-MV | | |
|---|---|---|---|---|---|---|
| | AUPRC | AUP | AUR | AUPRC | AUP | AUR |
| IG | 0.9282±0.0016 | 0.7775±0.0010 | 0.6926±0.0017 | 0.2369±0.0020 | 0.5150±0.0048 | 0.3211±0.0032 |
| Dynamask | 0.2290±0.0012 | 0.3422±0.0037 | 0.5170±0.0013 | 0.2836±0.0021 | 0.6369±0.0047 | 0.1816±0.0015 |
| WinIT | 0.4171±0.0016 | 0.5106±0.0026 | 0.3909±0.0017 | **0.3515**±0.0014 | **0.6547**±0.0026 | 0.3423±0.0021 |
| Ours | **0.9974**±0.0002 | **0.7964**±0.0009 | **0.8313**±0.0011 | 0.1298±0.0017 | 0.1307±0.0022 | **0.4751**±0.0015 |

Table 16: Explainer results with LSTM predictor on FreqShapes and SeqComb-MV synthetic datasets.

| Method | ECG | | |
|---|---|---|---|
| | AUPRC | AUP | AUR |
| IG | 0.5037±0.0018 | 0.6129±0.0026 | 0.4026±0.0015 |
| Dynamask | 0.3730±0.0012 | 0.6299±0.0030 | 0.1102±0.0007 |
| WinIT | 0.3628±0.0013 | 0.3805±0.0022 | 0.4055±0.0009 |
| Ours | **0.6057**±0.0018 | **0.6416**±0.0024 | **0.4436**±0.0017 |

Table 17: Explainer results with LSTM predictor on ECG dataset.

the best AUPRC on SeqComb-MV and ECG datasets. However, the performance for FreqShapes saturates at very high values for both TIMEX and IG, making the comparison more difficult for AUPRC. Tables 16,17 show the results of TIMEX against strong baselines with an LSTM predictor. TIMEX performs very well for both FreqShapes and ECG datasets, achieving the highest AUPRC, AUP, and AUR for both datasets. For SeqComb-MV, TIMEX did not converge. However, no explainer performed well for this task, achieving lower results than for the transformer and CNN predictors.

**Visualization under different architectures.** We add a visualization on the FreqShapes dataset with various explainers on different underlying architectures. Figure 7 shows this visualization. Explanations are similar across models. IG outputs appear similar, but TIMEX has a higher recall for essential patterns, which is reflected in quantitative results.

### C.9 TIMEX on diverse tasks and datasets

We experiment with TIMEX on various tasks and types of datasets to demonstrate the generality of the method.

**Irregular time series.** We experiment with TIMEX on irregular time series classification. We introduce an irregularly sampled version of SeqComb-MV, randomly dropping an average of 15% of time steps. This results in variable-length time series, addressing both W1 and W2. We then follow Zhang et al. [S23], using a time series transformer with an irregular attention mask. We train only with direct-value masking to avoid direct interference with this mechanism.

Table 20 shows the results of this experiment. We compare TIMEX to Integrated Gradients (IG) performance because, given the nuance of learning from irregularly-sampled datasets [S23], most

| Method | FreqShapes | | | SeqComb-MV | | |
|---|---|---|---|---|---|---|
| | AUPRC | AUP | AUR | AUPRC | AUP | AUR |
| IG | **0.9955**±0.0005 | **0.8754**±0.0008 | 0.7240±0.0015 | 0.5979±0.0027 | **0.8858**±0.0014 | 0.2294±0.0013 |
| Dynamask | 0.2574±0.0008 | 0.4432±0.0032 | 0.5257±0.0015 | 0.4550±0.0016 | 0.7308±0.0025 | 0.3135±0.0019 |
| WinIT | 0.5321±0.0018 | 0.6020±0.0025 | 0.3966±0.0017 | 0.5334±0.0011 | 0.8324±0.0020 | 0.2259±0.0020 |
| Ours | 0.9941±0.0002 | 0.6915±0.0010 | **0.8522**±0.0009 | **0.7016**±0.0019 | 0.7670±0.0012 | **0.4689**±0.0016 |

Table 18: Explainer results with CNN predictor on FreqShapes and SeqComb-MV synthetic datasets.

| Method | ECG | | |
|---|---|---|---|
| | AUPRC | AUP | AUR |
| IG | 0.4949±0.0010 | 0.5374±0.0012 | **0.5306**±0.0010 |
| Dynamask | 0.4598±0.0010 | 0.7216±0.0027 | 0.1314±0.0008 |
| WinIT | 0.3963±0.0011 | 0.3292±0.0020 | 0.3518±0.0012 |
| Ours | **0.7844**±0.0014 | **0.8706**±0.0012 | 0.3972±0.0010 |

Table 19: Explainer results with CNN predictor on ECG dataset.

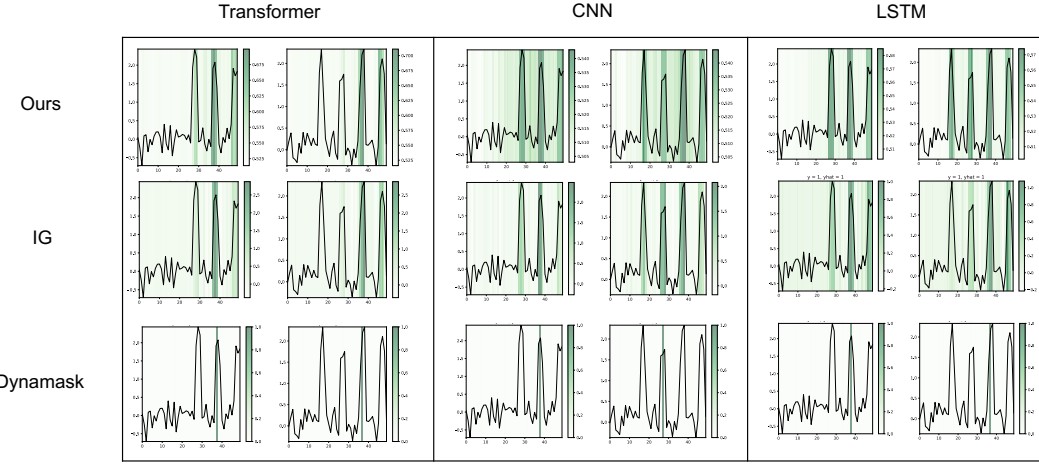

Figure 7: Visualization of FreqShapes dataset of TIMEX explainer (top row), integrated gradients (IG), and Dynamask explainers on the FreqShapes dataset with transformer, CNN, and LSTM architectures.

|       | AUPRC            | AUP              | AUR              |
| ----- | ---------------- | ---------------- | ---------------- |
| IG    | 0.286±0.004      | 0.617±0.009      | **0.372**±0.009  |
| TIMEX | **0.434**±0.007  | **0.666**±0.010  | 0.196±0.003      |

Table 20: Performance of TIMEX and IG on a modified version of the SeqComb-MV dataset such that the samples are irregularly sampled.

baselines do not apply to irregularly-sampled time series datasets without significant changes that are out-of-scope for our work. TIMEX demonstrates superior performance in this setting, outperforming IG by an over 1.5x improvement in AUPRC.

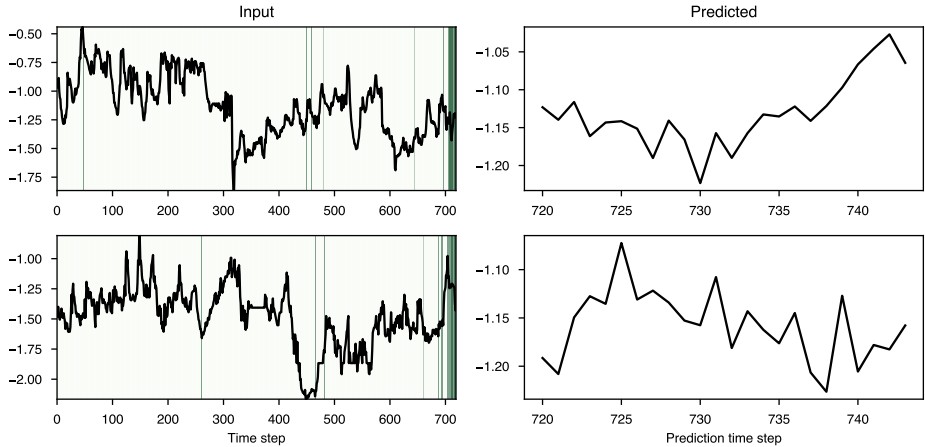

Figure 8: Forecasting experiment on the ETTh1 dataset with TIMEX explanations.

**Forecasting.** We demonstrate TIMEX 's generalizability to diverse tasks by explaining a forecasting model. We use the ETTh1 dataset [S24] and a vanilla forecasting transformer. To modify TIMEX for forecasting, we first extract embeddings used for MBC by max pooling over hidden states of the decoder. Next, we used a revised LC loss that used MSE as the distance function between predictions rather than JS divergence. We show visualizations of two samples in Figure 8. Explanations are in the left column, while forecasted time steps are in the right column. A few patterns emerge:

1. TIMEX identifies late time steps as crucial for the forecast, an expected result for a dataset with small temporal variance.

2. (a): the forecast is an increasing sequence, and TIMEX identifies a region of an expanding sequence (time 450-485). This suggests the model uses this increasing sequence as a reference for forecasting.

3. (b): a sharp upward spike is forecasted at the beginning of the predicted window. Correspondingly, TIMEX identifies a local minimum around time 260 and a sharp upward spike around time 475-490.

This preliminary experiment demonstrates that TIMEX shows the potential to extract meaningful explanations from forecasting datasets.

### C.10   Quantitative analysis of landmark explanations

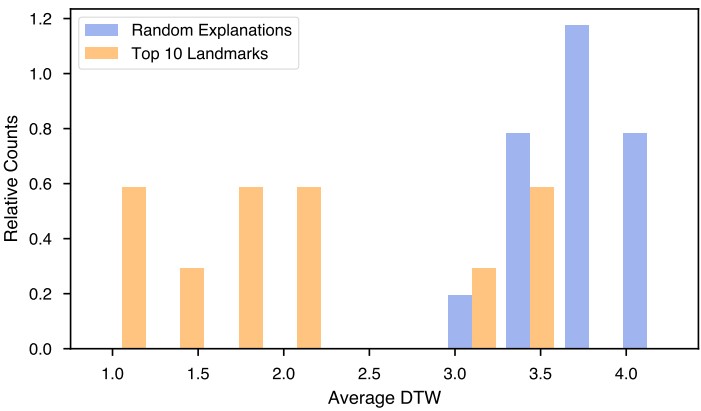

Figure 9: Distributions of DTW distance between structures of masked-in portions of explanations.

To examine landmark quality on the ECG dataset, we compare two groups: 1) the landmark group, with the five nearest-neighbors around the top 10 filtered landmarks, and 2) the random group, with five random explanations. We then compare the structure of the most salient values in the samples. We mask in the top-10 time steps as identified by TIMEX, then compute DTW [S25] distance between samples in each group. We then plot the distribution of average within-group DTW distances in Figure 9. A lower average DTW within their groups demonstrates the high quality of learned landmarks.

| Clustering method | NMI | ARI |
|---|---|---|
| Landmark | **0.191**±0.014 | **0.152**±0.026 |
| K-means | 0.147±0.010 | 0.025±0.001 |
| Random | 0.109±0.030 | 0.027±0.042 |

Table 21: Quantitative experiment comparing learned landmark explanations to K-means and Random clusters in the explanation latent space.

We have run an additional experiment comparing the clustering performance of our landmarks to those learned by k-means on the final latent space. On the ECG dataset, we match embeddings to their nearest landmarks, to their nearest k-means cluster centroids, and to randomly-selected points as a baseline. For each method, we use 50 clusters. We then evaluate each clustering method's Normalized Mutual Information (NMI) and Adjusted Rand Index (ARI), which are standard clustering metrics, against the ground-truth labels and report the standard error (+/-) computed over 5-fold cross-validation. Higher metrics for one set of centroids would indicate that proximity is more meaningful for prediction. The results are shown in Table 21. Here, we see higher NMI and ARI for landmarks, which means the landmarks are a better heuristic for forming clusters than the K-means centroids for this task.

# Appendix References

[S1] Penghang Yin, Jiancheng Lyu, Shuai Zhang, Stanley Osher, Yingyong Qi, and Jack Xin. Understanding straight-through estimator in training activation quantized neural nets. *arXiv preprint arXiv:1903.05662*, 2019.

[S2] Pengyu Cheng, Chang Liu, Chunyuan Li, Dinghan Shen, Ricardo Henao, and Lawrence Carin. Straight-through estimator as projected wasserstein gradient flow. *arXiv preprint arXiv:1910.02176*, 2019.

[S3] Veenu Rani, Syed Tufael Nabi, Munish Kumar, Ajay Mittal, and Krishan Kumar. Self-supervised learning: A succinct review. *Archives of Computational Methods in Engineering*, pages 1–15, 2023.

[S4] Alec Radford, Jong Wook Kim, Chris Hallacy, Aditya Ramesh, Gabriel Goh, Sandhini Agarwal, Girish Sastry, Amanda Askell, Pamela Mishkin, Jack Clark, et al. Learning transferable visual models from natural language supervision. In *International conference on machine learning*, pages 8748–8763. PMLR, 2021.

[S5] Ting Chen, Simon Kornblith, Mohammad Norouzi, and Geoffrey Hinton. A simple framework for contrastive learning of visual representations. In *International conference on machine learning*, pages 1597–1607. PMLR, 2020.

[S6] Kaiming He, Haoqi Fan, Yuxin Wu, Saining Xie, and Ross Girshick. Momentum contrast for unsupervised visual representation learning. In *Proceedings of the IEEE/CVF conference on computer vision and pattern recognition*, pages 9729–9738, 2020.

[S7] Xun Wang, Xintong Han, Weilin Huang, Dengke Dong, and Matthew R Scott. Multi-similarity loss with general pair weighting for deep metric learning. In *Proceedings of the IEEE/CVF conference on computer vision and pattern recognition*, pages 5022–5030, 2019.

[S8] Debidatta Dwibedi, Yusuf Aytar, Jonathan Tompson, Pierre Sermanet, and Andrew Zisserman. Temporal cycle-consistency learning. In *Proceedings of the IEEE/CVF conference on computer vision and pattern recognition*, pages 1801–1810, 2019.

[S9] Vipin Pillai, Soroush Abbasi Koohpayegani, Ashley Ouligian, Dennis Fong, and Hamed Pirsiavash. Consistent explanations by contrastive learning. In *Proceedings of the IEEE/CVF Conference on Computer Vision and Pattern Recognition*, pages 10213–10222, 2022.

[S10] Tao Han, Wei-Wei Tu, and Yu-Feng Li. Explanation consistency training: Facilitating consistency-based semi-supervised learning with interpretability. In *Proceedings of the AAAI conference on artificial intelligence*, volume 35, pages 7639–7646, 2021.

[S11] Hao Guo, Kang Zheng, Xiaochuan Fan, Hongkai Yu, and Song Wang. Visual attention consistency under image transforms for multi-label image classification. In *Proceedings of the IEEE/CVF conference on computer vision and pattern recognition*, pages 729–739, 2019.

[S12] Matthew Watson, Bashar Awwad Shiekh Hasan, and Noura Al Moubayed. Using model explanations to guide deep learning models towards consistent explanations for ehr data. *Scientific Reports*, 12(1):19899, 2022.

[S13] Sanjoy Dasgupta, Nave Frost, and Michal Moshkovitz. Framework for evaluating faithfulness of local explanations. In *International Conference on Machine Learning*, pages 4794–4815. PMLR, 2022.

[S14] Huanlai Xing, Zhiwen Xiao, Dawei Zhan, Shouxi Luo, Penglin Dai, and Ke Li. Selfmatch: Robust semisupervised time-series classification with self-distillation. *International Journal of Intelligent Systems*, 37(11):8583–8610, 2022.

[S15] Lujun Li and Zhe Jin. Shadow knowledge distillation: Bridging offline and online knowledge transfer. *Advances in Neural Information Processing Systems*, 35:635–649, 2022.

[S16] Defang Chen, Jian-Ping Mei, Hailin Zhang, Can Wang, Yan Feng, and Chun Chen. Knowledge distillation with the reused teacher classifier. In *Proceedings of the IEEE/CVF conference on computer vision and pattern recognition*, pages 11933–11942, 2022.

[S17] Robert Geirhos, Jörn-Henrik Jacobsen, Claudio Michaelis, Richard Zemel, Wieland Brendel, Matthias Bethge, and Felix A. Wichmann. Shortcut learning in deep neural networks. *Nature Machine Intelligence*, 2(11):665–673, Nov 2020.

[S18] Lukas Faber, Amin K. Moghaddam, and Roger Wattenhofer. When comparing to ground truth is wrong: On evaluating gnn explanation methods. In *Proceedings of the 27th ACM SIGKDD Conference on Knowledge Discovery & Data Mining*, pages 332–341, 2021.

[S19] Borys Surawicz, Rory Childers, Barbara J Deal, and Leonard S Gettes. Aha/accf/hrs recommendations for the standardization and interpretation of the electrocardiogram: part iii: intraventricular conduction disturbances: a scientific statement from the american heart association electrocardiography and arrhythmias committee, council on clinical cardiology; the american college of cardiology foundation; and the heart rhythm society: endorsed by the international society for computerized electrocardiology. *Circulation*, 119(10):e235–e240, 2009.

[S20] Mariana Floria, Alexandra Noela Parteni, Ioana Alexandra Neagu, Radu Andy Sascau, Cristian Statescu, and Daniela Maria Tanase. Incomplete right bundle branch block: Challenges in electrocardiogram diagnosis. *Anatolian Journal of Cardiology*, 25(6):380, 2021.

[S21] Chirag Agarwal, Eshika Saxena, Satyapriya Krishna, Martin Pawelczyk, Nari Johnson, Isha Puri, Marinka Zitnik, and Himabindu Lakkaraju. Openxai: Towards a transparent evaluation of model explanations. *arXiv preprint arXiv:2206.11104*, 2022.

[S22] Narine Kokhlikyan, Vivek Miglani, Miguel Martin, Edward Wang, Bilal Alsallakh, Jonathan Reynolds, Alexander Melnikov, Natalia Kliushkina, Carlos Araya, Siqi Yan, and Orion Reblitz-Richardson. Captum: A unified and generic model interpretability library for pytorch, 2020.

[S23] Xiang Zhang, Marko Zeman, Theodoros Tsiligkaridis, and Marinka Zitnik. Graph-guided network for irregularly sampled multivariate time series. In *International Conference on Learning Representations, ICLR*, 2022.

[S24] Haoyi Zhou, Shanghang Zhang, Jieqi Peng, Shuai Zhang, Jianxin Li, Hui Xiong, and Wancai Zhang. Informer: Beyond efficient transformer for long sequence time-series forecasting. In *Proceedings of the AAAI conference on artificial intelligence*, volume 35, pages 11106–11115, 2021.

[S25] Meinard Müller. Dynamic time warping. *Information retrieval for music and motion*, pages 69–84, 2007.

