# OpenReview forum: "Encoding Time-Series Explanations through Self-Supervised Model Behavior Consistency"
_NeurIPS.cc/2023/Conference — NeurIPS 2023 spotlight_

### Official Review · Reviewer_wJLT · 2023-06-19

**Soundness:** 2 fair
**Presentation:** 2 fair
**Contribution:** 3 good
**Rating:** 5
**Confidence:** 2

**Summary:**

This paper introduces a novel method to train an interpretable surrogate of pre-trained  time series classification models. TimeX produces a latent embedding of time series observation and outputs classification probabilities that are both consistent with a reference model. At the same time, it identifies time localized patterns in the time series that allowed it to make predictions. Last, it learns landmarks in the latent space that allow the user to easily compare the observed time series.

From pre-trained encoder $G$ and decoder $F$, TimeX trains an explanation generator $H_E$, encoder $G_E$ and decoder $F_E$. $H_E$ produces a discrete masking function that allows to localize and consider only sparse yet connected regions of interest in the input time series. $G_E$ and  $F_E$ are jointly trained to respectively preserve the "topology" of the latent spaces and the output distributions.

Please note that the submission is not in my area of expertise. I can assess the soundness of the claims and the presentation but it is hard to evaluate its broader contribution to the field.

**Strengths:**

The notion of Model Behavior Consistency is, from my understanding, the main contribution of this paper. Although many methods aim at  embedding high dimensional observations in topology-preserving latent spaces, the idea of learning an interpretable embedding consistent with pre-trained models is novel to my knowledge. I find the approach particularly relevant as high-capacity models become more and more accessible.

Although the technical description of the method is not very well written (see below), each modeling choice is convincing and well motivated (each term of the *ad-hoc* loss is easily interpretable and the need for discrete masking for time series is well demonstrated).

The experimental results demonstrate improved performance that I cannot really evaluate (see above).





**Weaknesses:**

1) Some statements seem inaccurate if not incorrect. (I will be happy to be proven wrong, as I am not an expert in Time-Series Classification)

   - line.22 "State-of-the-art time series models are high-capacity pre-trained neural networks" -> This might be true for time-series classification, but seems wrong in other areas like forecasting, where uncertainty quantification is key.

   - There is an apparent conflict between the need for "pinpointing the specific location [...] that influence the model's prediction" (line 24) and the fact that "temporal patterns only show up when looking at time segments and long term trends" (line 54). For example, if the feature of interest is the frequency content of a time series, it seems pointless to precisely localize it in time. From my understanding, TimeX reaches an in-between suitable to pinpoint patterns "broadly-localized" in time. The paper would gain in clarity by stating more clearly what the method is trying to achieve.

   - line 47. "the uninterpretable nature of time series data". I would avoid such vague statements. For example, the QRS intervals subsequently analyzed are straightforwardly interpretable and corresponds to ventricles depolarization and heart contraction.

2) The description of the method is not intelligible but could easily be improved.
   - line 154. The functions $D_Z$ and $D_Z^E$ are used to motivate Model Behavior Consistency, but it is not explained that they convey a notion of distance in the latent space (this only shows up line 241)

   - Although I understand  the relationship $D_Z(z_i, z_j) \approx D_Z^E(z_i^E, z_j^E)$ (line 154) and its implication, it is very confusing to write $z_j^E = G^E(E(x_i))$ (line 153).  $z_j^E$ is a latent representation of the masked time series $x^m_j$. And $E$ is defined above as explanation that takes value in $[0, 1]$ (see line 189). Did I miss something ?

   - The notation $H^E: \mathcal{X} \rightarrow p \in [0, 1]^{T \times d}$ does not make sense (line 184). Is $\mathcal{X}$ the input space of the function $H^E$ or its input ? In the former case, one should write $H^E: \mathcal{X} \rightarrow [0, 1]^{T \times d}$. In the latter case it implies that the mask applies uniformly to the whole dataset $\mathcal{X} = (x_i)_i$ which contradicts my intuition from Figure 2.

3) Figure 4 clearly illustrates what part of the ECG the model based its prediction on. Nevertheless, there is no further analysis of the discovered temporal patterns. Are they unexpected or are they stereotyped signals easily classifiable by a physician ? In the latter case, it would have been interesting to use an additional classification task and illustrate how TimeX can identify new temporal patterns in observations, and possibly be used for exploratory data analysis.

Minor points: line 91. XAI: abrevation not defined

**Questions:**

1) I did not understand how and exactly when the Landmarks are learned. Is it post-hoc ? Or is there some partial overlap with the training of the rest of the model which encourages the finding a parsimonious latent embedding ?

2) It is not really discussed how the two latent spaces dimensions are related. Since $G^E$ is initialized with $G$, I am assuming that they are the same, but I imagine that one might use a lower dimensional embedding space of explanation. It would be relatively straightforward to initialize $G^E:= P G$ where $P$ is a linear projection on a lower dimensional space. Would it make the model even more interpretable (and avoid post-hoc treatment with UMAP as in Figure 2) or would it penalize the model too much ?

**Limitations:**

Authors adequately addressed the limitations of their model.

---

> ### Author Rebuttal · Authors · 2023-08-09
>
> Thank you for providing valuable comments and critiques of our work. We have worked hard to improve the communication of our method, and we kindly ask you to raise your score. Please reach out to us with any questions.
> ### W1a: State-of-the-art models
> Thank you for pointing out this misleading statement. Many variants of transformers, often very large models, have become preeminent models for various time series tasks. See recent works [1,2] for time series classification, [3,4] for forecasting, and [5] for anomaly detection. Additionally, uncertainty quantification methods have been developed for large neural networks used in forecasting [6]. We have revised our statement: “Prevailing time series models are often high-capacity pre-trained neural networks.”
> ### W1b: Modeling frequency content of time series
> There is indeed value in identifying the specific location of a temporal pattern. For example, in physiological signal diagnosis via ECG or EEG, the exact time that a pattern occurred is of interest to practitioners [7]. TimeX is highly-effective in this setting, identifying the location of a temporal pattern while providing similar explanations through the learned landmarks.
>
> However, predictions in time series can be driven by a variety of patterns, such as from the frequency domain. In these settings, TimeX’s explanations would be suboptimal as it cannot provide frequency-level explanations. However, baselines time series explainers (Dynamask and WinIT) would also fail in this case. While many time series predictors model the frequency domain [3], to the knowledge of the authors, no time series explainers have been developed with this capability. This desideratum is left for future work
> ### W1c: Uninterpretability of time series
> Time series data can be visually uninterpretable as in noisy samples and long sequences can obstruct visualization of meaningful signals. Thus, one saliency map is not enough for interpreting time series data, unlike computer vision and natural language processing where explainers rely on human perception by overlaying saliency maps on images or tokens. This motivates landmark explanations to provide a mechanism to relate instances on the dataset. We agree that QRS intervals are indeed interpretable by experts, but this is a special case. Decades of research has focused on understanding ECG due to its importance in medicine, thus these data are readily interpretable.
>
> We have edited this claim: “While explainers designed for other modalities can be adapted to time series, their inherent biases can miss important structures in time series and their reliance on isolated visual interpretability do not translate effectively to the time series where data are less-immediately interpretable.”
> ### W2a: D_Z and D_{Z^E}:
> We have now moved our initial definition of $D_Z$ and $D_{Z^E}$ to Section 3 to make the problem formulation more readable.
> ### W2b: Notation of G^E:
> We agree it is an abuse of notation to use $E$ to represent both the explanation and the masked version of the input sample. We have edited the text to use $x_j^m$ to denote the input to $G^E$.
> ### W2c: Notation of H^E:
> Thank you for pointing this out, $H^E: \mathcal{X} \rightarrow [0,1]^{T \times d}$ is the proper notation. We have made this edit in text.
> ### W3: Discovered temporal patterns
> The landmark explanations group together similar explanations that constitute temporal patterns across the dataset. It would be ideal to consult domain experts and conduct a user study to understand if the landmarks represent novel patterns. However, this cannot be conducted within the time allotted for rebuttals. Therefore, we provide a quantitative analysis in **Figure 4**.
>
> To examine landmark quality on the ECG dataset, we compare two groups: 1) the landmark group, with the five nearest-neighbors around the top-10 filtered landmarks and 2) the random group, with five random explanations. We then compare the structure of the most salient values in the samples. We mask-in the top-10 time steps as identified by TimeX, and then compute DTW [8] distance between samples in each group. We then plot the distribution of average within-group DTW distances in **Figure R4**. The high quality of learned landmarks is demonstrated by a lower average DTW within their groups. We have added this analysis to the Appendix.
> ### Q1: How landmarks are learned:
> Landmarks are learned during training. We do not backpropagate the gradients of the landmark explanation learning to the entirety of the TimeX model to prevent the landmark explanations from harming the performance of the explainer.
> ### Q2: Latent spaces:
> $G$ and $G^E$ embed $Z$ and $Z^E$ respectively, but the inputs to both $G$ and $G^E$ are different—$\mathcal{X}$ and $\mathcal{X}^m$, respectively—as described in Section 4.2. This allows TimeX to optimize the mask $M_{\mathcal{X}}$ on $G^E$ rather than $G$, where $\mathcal{X}^m$ might be out-of-distribution from $\mathcal{X}$. Therefore, the formulation of $G^E := PG$ is too strict and might prevent the model from learning complex temporal patterns.
>
> **References**: [1] Zerveas et.al., A transformer-based framework for multivariate time series representation learning. KDD 2021.
> [2] Chowdhury et.al., TARNet: Taskaware reconstruction for time-series transformer. KDD 2022.
> [3] Zhou et.al., FEDformer: Frequency enhanced decomposed transformer for long-term series forecasting. ICML 2022.
> [4] Wu et.al., Autoformer: Decomposition Transformers with Auto-Correlation for Long-Term Series Forecasting. NeurIPS 2021.
> [5] Tuli et.al., TranAD: Deep transformer networks for anomaly detection in multivariate time series data. VLDB 2022.
> [6] Wu et.al., Quantifying Uncertainty in Deep Spatiotemporal Forecasting. KDD 2021.
> [7] Lundberg et al., Explainable machine-learning predictions for the prevention of hypoxaemia during surgery. *Nat. Bio. Eng.* 2018.
> [8] Muller, Dynamic time warping. Information retrieval for music and motion 2007.

---

> > ### Comment · Reviewer_wJLT · 2023-08-14
> >
> > I thank the authors for their response. Their clarifications were insightful and important (at least to me) for the understanding of their paper.
> >
> > Nevertheless, I still find it disappointing that although landmarks inference is presented as a key feature of the method, the paper does use them in any way.
> >
> > I did not doubt that the landmarks were quantitatively good descriptors of the data (and the authors actually provided further evidence for that), but as they pointed out,  "it would be ideal to consult domain experts and conduct a user study to understand if the landmarks represent novel patterns" or at least meaningful. It may be worth showing, at least, that the Landmarks are more stable, or more "interpretable" than simply applying K-Means on the latent space after fitting TimeX.

---

> > > ### Author Response · Authors · 2023-08-18
> > >
> > > Thanks for your continued feedback, we’re glad to hear that our clarifications were helpful and that your remaining concern is just with the landmarks.
> > >
> > > Our main contribution is the TimeX method, not landmarks. As you note in your initial review, TimeX is highly-relevant, and our modeling choices are well-motivated. Further, our main results, presented in Tables 1, 2, and 3, along with Figure 4, all use the most-important metrics in our field. While conducting human studies is an exciting future step, it is out of scope for this work. Our intention is not to present landmarks as a key feature; we will address your comments by toning down our language and expanding the limitation section in the main manuscript.
> > >
> > > Still, we address your lingering concerns about interpreting landmarks in 3 points:
> > > 1. Designing, conducting, and evaluating high-quality user studies is a major undertaking worthy of its own complete work. Doing these studies for explainability is important work, and it’s an active area of research on its own. Given that TimeX consistently outperforms state-of-the-art explainability methods, we hope it will be included in such studies.
> > > 2. We have now conducted a preliminary discussion with a medical doctor who frequently works with ECG data and with machine learning. In this conversation, the doctor said there is likely meaning in the landmarks in Figure 4: Series 1, 2, and 3 clearly have irregular beats, possibly related to long QT periods. 4, 5, and 6 are heterogenous, possibly related to variance in the ST segment. A more comprehensive user study done in collaboration with medical experts would, however, require considerably more time than what is possible in this relatively short discussion period. To address your question, we will make sure to clearly discuss user studies as future work and include references to sample human evaluation studies.
> > > 3. We have run an additional experiment comparing the clustering performance of our landmarks to those learned by k-means on the final latent space. On the ECG dataset, we match embeddings to their nearest landmarks, to their nearest k-means cluster centroids, and to randomly-selected points as a baseline. For each method, we use 50 clusters. We then evaluate each clustering method's Normalized Mutual Information (NMI) and Adjusted Rand Index (ARI), which are standard clustering metrics, against the ground-truth labels and report standard error (+/-) computed over 5-fold cross validation. Higher metrics for one set of centroids would indicate that proximity is more meaningful with respect to the prediction task. The results are shown in the table below. Here we see higher NMI and ARI for landmarks, which means the landmarks are a better heuristic for forming clusters than the K-means centroids for this task.
> > >
> > > | Clustering Method | NMI | ARI |
> > > | :----: | :----: | :----: |
> > > | Landmarks | 0.191 +/-  0.014 | 0.152 +/-  0.026 |
> > > | K-means | 0.147 +/- 0.010 | 0.025 +/- 0.001 |
> > > | Random | 0.109 +/- 0.030 | 0.027 +/- 0.042 |

---

### Official Review · Reviewer_9pJd · 2023-07-03

**Soundness:** 4 excellent
**Presentation:** 3 good
**Contribution:** 3 good
**Rating:** 7
**Confidence:** 2

**Summary:**

This paper proposed TimeX that creates an interpretable surrogate model for pretrained models. To ensure faithfulness to the reference model, this paper introduces a self-supervised objective, model behavior consistency, a novel formulation that ensures the preservation of relationships in the latent space induced by the pretrained model, as well as the latent space induced by TimeX. As a result, TimeX could find well-distributed landmarks and highlight a range of salient timestamps.


**Strengths:**

1. It is well-written and has strong motivation.
2. This is the first paper to suggest in-hoc explanations for time series data.
3. This article offers additional justifications for its design decisions, including discrete masking and consistency learning.
4. The evaluation is very convincing, using diverse datasets and many compared methods.

**Weaknesses:**

1. No discussion about the quality of explanation. Please see questions.


**Questions:**

1. Any metric for the quality of the explanation? For example, better explanation should show high IoU with ground truth. This kind of qualitative analysis would assure the superiority of TimeX.
2. If Table 3 is at the top of the page, the visibility would be improved.



**Limitations:**

The limitation that authors provided is acceptable as there is no ground truth for other tasks except classification.

---

> ### Author Rebuttal · Authors · 2023-08-09
>
> Thank you for the useful feedback about our work. We appreciate the reviewer’s acknowledgement of the novelty of our work, as well as the utility and diversity of our extensive evaluations. We have responded to your concerns in the below comments, running 1 additional experiment. We encourage you to reach out if there are any further questions or clarifications needed.
>
> ### W1 and Q1: Quality of explanations
> We believe there is some confusion regarding our experiments. We use two primary setups to evaluate the quality of TimeX explanations - ground-truth comparison and ablation analysis. These are both common analyses in XAI literature, including time series XAI literature [1,2,3].
>
> In the ground-truth comparison, we compare the explanations of TimeX and baselines to known temporal signals in each dataset that are known to drive the model prediction. We use four synthetic datasets and one real-world dataset (ECG) for this evaluation setup. We then use three separate metrics that compare **ground-truth explanations to explanations from each XAI method**. The area under the precision-recall curve (AUPRC), area under precision curve (AUP), and area under recall curve (AUR) are all metrics to evaluate the correspondence between explanations from each method and ground-truth explanations. These evaluations are described in detail throughout Section 5, 6, and Appendix C. Intersection over Union (IoU) measures the same qualities of explanations—correspondence to ground-truth explanations—as AUPRC, AUP, and AUR. Therefore, our chosen metrics in this setup measure the same qualities of explanations as the IoU metric.
>
> | Method | IoU | AUPRC |
> |---|---|---|
> | Integrated Gradients | 0.3750 $\pm$ 0.0022 | 0.5760 $\pm$ 0.0022 |
> | Dynamask | 0.2958 $\pm$ 0.0014 | 0.4421 $\pm$ 0.0016 |
> | TimeX | 0.5214 $\pm$ 0.0019 | 0.7124 $\pm$ 0.0017 |
>
>
>
> To illustrate this point, we include results in the table above of TimeX and two of the strongest baselines—Dynamask and Integrated Gradients—measuring the IoU score on the SeqCombSingle dataset. For comparison to our metrics, we include the AUPRC results for the same methods. The IoU metric has high correlation with the AUPRC metric, with each metric resulting in the same ranking of methods and TimeX achieving the highest metric.
>
> The other setup we use to evaluate explanation quality is an occlusion analysis. In this case, we mask-out (or occlude) values predicted by the explainer to be un-important, and we measure the drop in classification performance for the predictor under these explanation-dependent perturbations. This allows us to evaluate on real-world datasets where no ground-truth explanation is available. We describe these experiments in Section 6, R2 and results are shown in in-text Figure 3.
>
> We hope that this clears up some confusion about our evaluations. All of our experiments seek to evaluate the quality of explanations from each method. We have extensive evaluations in this regard, using 8 datasets with a combination of synthetic and real-world time series datasets. In addition, our Appendix includes additional analyses on a variety of time series classification architectures. We have included a brief experiment using IoU metrics to illustrate the point that our experiments capture similar qualities to that measured by the IoU score.
>
> ### Q2: Table 3
> Thank you for this suggestion concerning the readability of our manuscript. In response, we have moved Table 3 to the top of the page.
>
> **References**:
> [1]: Agarwal et al., “Evaluating explainability for graph neural networks”, *Scientific Data*, 2023.
> [2]: Crabbe and van der Schaar, “Explaining Time Series Predictions with Dynamic Masks,” ICML 2021.
> [3]: Tonekaboni et al., “What went wrong and when? Instance-wise Feature Importance for Time-series Models”, NeurIPS 2019.

---

> > ### Comment · Reviewer_9pJd · 2023-08-13
> > **Response to the Rebuttal**
> >
> > I appreciate detailed reply for my review. I do not have any question at this point.

---

### Official Review · Reviewer_7JHq · 2023-07-04

**Soundness:** 4 excellent
**Presentation:** 4 excellent
**Contribution:** 4 excellent
**Rating:** 7
**Confidence:** 4

**Summary:**

This paper presents an in-hoc interpretability mechanism to explain time series prediction. In particular, the authors train an interpretable surrogate model by learning H^E and G^E in the embedding space.

The objective function optimizes model behavior consistency by considering the distance in the training embedding, Z, and explained embedding, Z^E.


**Strengths:**

1. Discrete masking as opposed to continuous masking
2. Landmark to partition latent space visualization and shape of the signal
3. Can be generalized to other classification tasks and neural model architectures


**Weaknesses:**

As demonstrated by the ablation studies, the label alignment (LA) loss is always better than MBC in isolation.



**Questions:**

Table 12 results in the Supplementary materials require an explanation of the SeqComb-MV dataset. Specifically, the  MBC and LA loss in isolation has very poor AUP, ~0.0576. But the combined MBC + LA results in AUP 0.8326. The performance boost is really large.

Do the authors have any explanation for this behavior?


**Limitations:**

Yes, the authors explained use cases where their model is not the best.

---

> ### Author Rebuttal · Authors · 2023-08-08
>
> We are very grateful for your constructive feedback about our work. We appreciate that you recognized the core contributions of our work, as well as the novelty of multiple components of TimeX, such as discrete masking, landmark explanations, and model behavior consistency learning. We respond to your concerns about our work in the following comment. Please reach out if you have any additional questions or clarifications about our work.
>
> ### W1 and Q1: LA vs. MBC Loss in Isolation
> We thank you for noting this peculiar result in our work. The explanation for this behavior gets at a core of the motivation behind TimeX: faithful explainers should match multiple internal states of the model.
>
> First, we recall an argument presented in Section 4.3, where we justify MBC. We remark that perturbation-based methods have a similar idea to TimeX: find some sparse perturbation to the input that can preserve the output state of the model. This is often done through observing the predicted label from the model under an applied mask or perturbation, e.g., in one of our baselines, Dynamask [1]. A perturbation that preserves the output state is said to be “faithful” to the model because it is assumed that the perturbation causes. In a sense, MBC generalizes this idea to latent spaces, ensuring that invariances are preserved on the latent space of the model as well as the prediction space.
>
> Beyond the introduction of MBC alone, another core contribution of our work focuses on optimizing faithfulness to predictor models on multiple levels. We use multiple hidden or output states of the model, e.g., a latent space and a logit space, on which the explainable surrogate should match the reference predictor. The hypothesis behind this innovation is that model behavior (the exact objective we are trying to explain) cannot be fully captured by one state, e.g., a predicted label, but rather by multiple states throughout the model. A similar assumption is made in knowledge distillation, where methods often optimize student models to match the teacher on multiple layers of the network. **Therefore, MBC and LA together enforce adherence to model behavior on two fronts: the latent space and prediction space, respectively.** This explains the behavior that you mention in our ablation studies: MBC and LA perform poorly alone, but together, these two losses provide a powerful objective to train the model.
>
> **References**:
> [1] Crabbe and van der Schaar, “Explaining Time Series Predictions with Dynamic Masks,” ICML 2021.

---

> > ### Comment · Reviewer_7JHq · 2023-08-20
> >
> > I thank the authors for their detailed explanation. I do not have further queries at this point.

---

### Official Review · Reviewer_Mf92 · 2023-07-09

**Soundness:** 3 good
**Presentation:** 3 good
**Contribution:** 3 good
**Rating:** 6
**Confidence:** 2

**Summary:**

TimeX proposes a explanation module that is trained along-side the model to provide more consistent explainiations. This is done by making the internal embeddings of explaination modules consistent with that of full model such as distance consistency and label consistency.

**Strengths:**

1. The methodology is well motivated and presented.
2. The consistency losses encourage expaliation module features to learn from latent space of the model.
3. The synthetic experiments are well setup and model provides significant performance improvements in synthetic and real-world dataset.

**Weaknesses:**

1. The novelty of the method seems unclear. Model consistency has been previously explored in neural explainations and this work dovtails to time-series models.
2.What is the additional computational complexity or resources required to train the explainer modules? How does it compare to other methods?

**Questions:**

See weaknesses

**Limitations:**

The paper presents a model behaviour consistency based method of generate explainations for time-series models. While the results are very significant, computational requirements relative to other post-hoc methods and the performance vs compute tradeoff is not clear.

---

> ### Author Rebuttal · Authors · 2023-08-08
>
> Thank you for your valuable feedback and criticism about our work. We greatly appreciate your claims that our work is “well motivated and presented” and that our explainer provides “significant performance improvements” over baseline explainers. We address concerns about novelty and computational efficiency in this response. If you feel our comments address your concerns, we kindly ask that you raise your score. Please reach out during the author-reviewer discussion if any questions are left unanswered.
>
> ### W1: Novelty
> Thank you for this comment. Our method is novel in its formulation, specifically one of our central contributions, model behavior consistency (MBC), is a novel formulation for training surrogate explainers for neural networks in time series. MBC states that the surrogate model should match the original time series model through mimicking its latent space and predictions. This is a novel extension of previous perturbation-based methods which seek to find some perturbation on the original input that preserves the prediction of the model. Instead of only preserving the predictions of the original model, TimeX also seeks to preserve the structure of the original model’s latent space, which increases faithfulness to original model behavior. See Section 4.3 for a more formal justification of MBC. While consistency has been explored in previous XAI works, we are the first to consider consistency between the predictor model and the explainable surrogate model. To draw this distinction, we contrast our method to two notions of consistency: 1) **consistency between explanations** and 2) **consistency as an explainability metric**.
>
> **Consistency between explanations**: This notion has been introduced in previous works in explainability literature. Pillai et al. [1] train a saliency explainer via contrastive learning that preserves consistency across the saliency maps for augmented versions of images. A few other works have explored maintaining consistency of explanations across various perturbations and augmentations, specifically in computer vision [2,3]. In one of the only previous works to consider explanation consistency in time series, Watson et al. [4] train an explainer on an ensemble of classifiers to optimize the consistency of explanations generated by an explainer applied to each individual classifier. TimeX does not seek to optimize simply consistency between explanations but rather consistency to the predictor model on which it is explaining.
>
> **Consistency as an explainability metric**: Dasgupta et al. [5] defines explanation consistency as similar explanations corresponding to similar predictions; this metric is then used as a proxy to faithfulness to evaluate the quality of explainers. However, Dasgupta et al. uses the notion of consistency to evaluate explainers, not to train and design a new explainer method. TimeX uses consistency as a learning objective rather than simply a metric.
>
> Our work differs from these previous formulations of explanation consistency. We seek to optimize the consistency not between explanations directly, as mentioned in previous works, but rather between the explainer and the model it is tasked with explaining. MBC attempts to ensure that the behavior of the explainable surrogate matches that of the original model. The definition of consistency in Dasgupta et al. is the closest to our definition of MBC; however, Dasgupta et al. does not seek to optimize the consistency of explainers but rather to evaluate the output of post-hoc explainers. TimeX directly optimizes the consistency between the surrogate model and the original predictor through the MBC loss, a novel formulation that seeks to increase the faithfulness of explanations generated by TimeX. Thus, we respectfully disagree with the sentence in your summary, which states that TimeX is trained to “provide more consistent explainiations [*sic*].” We have added these citations and arguments to the Appendix to make our distinction from previous work clearer.
>
> ### W2: Computational resources
> Please reference the general response Point 3 for discussion on this point. We sincerely thank you for requesting this analysis as this enriches the evaluation for TimeX.
>
> **References**:
> [1] Pillai et al., “Consistent Explanations by Contrastive Learning”, CVPR 2021.
> [2] Han et al., “Explanation Consistency Training: Facilitating Consistency-Based Semi-Supervised Learning with Interpretability”, AAAI 2021.
> [3] Guo et al., “Visual attention consistency under image transforms for multi-label image classification”, CVPR 2019.
> [4] Watson et al., “Using model explanations to guide deep learning models towards consistent explanations for EHR data”, Scientific Reports 2022.
> [5] Dasgupta et al., “Framework for Evaluating Faithfulness of Local Explanations”, ICML 2022.

---

> > ### Author Response · Authors · 2023-08-18
> > **Following up on our rebuttal**
> >
> > Dear Reviewer Mf92,
> >
> > Thank you again for your valuable feedback on our manuscript. We wanted to inquire if you had any additional comments or concerns regarding our response. We have carefully considered your feedback and have made the requested clarifications to improve the quality of the paper.
> >
> > Since it comes to the end of the author-reviewer discussion period, we would be grateful if you could let us know if the revisions have addressed your concerns or if there are any remaining issues that need to be addressed.
> >
> > Thank you for your attention to this matter. We look forward to hearing from you soon.
> >
> > Best regards,
> > Authors

---

### Official Review · Reviewer_Uw18 · 2023-07-27

**Soundness:** 4 excellent
**Presentation:** 4 excellent
**Contribution:** 3 good
**Rating:** 7
**Confidence:** 4

**Summary:**

The paper introduces a novel time series interpretability model called TIMEX. The challenge of interpreting time series models arises from the need to identify both the specific time series signals influencing predictions and their alignment with interpretable temporal patterns. TIMEX addresses the issue of model faithfulness by introducing model behavior consistency, ensuring relations in the latent space induced by the pretrained model are preserved when compared to relations in the TIMEX-induced latent space.

TIMEX provides discrete attribution maps, giving interpretable explanations for the model's predictions. Unlike existing methods, TIMEX goes further by learning a latent space of explanations, enabling visual aggregation of similar explanations and recognition of temporal patterns.

The evaluation of TIMEX on 4 synthetic and 4 real-world datasets, which includes case studies involving physiological time series, demonstrates its superior performance compared to state-of-the-art interpretability methods. TIMEX's innovative components hold promise for training interpretable models that capture the behavior of pretrained time series models.

**Strengths:**

Overall the proposed method is novel, effective, and well-evaluated. The main strengths of the paper are as follows:

1. The paper proposes a novel method for explaining time series models that is based on self-supervised learning. This approach has several advantages over traditional explanation methods, such as being able to learn explanations for models with complex temporal dynamics. The paper provides a clear and concise overview of the problem of explaining time series models and provides a detailed description of the TimeX method, including the rationale behind the design choices.

2. The authors presents a comprehensive evaluation of TimeX on a variety of experiments with time series models and datasets. The results show that TimeX is able to provide interpretable explanations that are both faithful to the model's predictions and informative about the underlying temporal patterns.

3. The paper also includes a number of case studies that demonstrate the use of TimeX to explain the predictions of time series models in real-world applications. These experiments help highlighting TimeX's capability to be used for gaining insights into the behavior of time series models and to identify potential problems with the models.

4. The authors evaluate TimeX through an extensive set of ablations and on a variety of time series models. TimeX shows to be effective in explaining a variety of time series models, including LSTM, CNN, and vanilla-Transformer model. This suggests that TimeX is a general-purpose method for explaining time series models.

**Weaknesses:**

The are a few weaknesses that if addressed or identified as a limitation for future can improve the paper:

1. As the current setup is presented, the model does not handle variable length time-series which is a common occurrence in medical time-series data. This could prove to be challenging to be extended from the current state of TimeX as aligning different time-series that have variable length would also become important.

2. Another common challenge of dealing with time-series data that is not addressed in the paper, is handling irregular time-step intervals. A discussion of this important limitation is missing in the paper. Irregularly sampled time-series is predominantly present in healthcare data that this paper focuses on and is important weakness of the TimeX that is not mentioned in the paper.

3. The method has not been evaluated on a wide range of different tasks. The authors of the paper evaluated TimeX on a variety of time series models and datasets, but they did not evaluate TimeX on a wide range of tasks. This means that it is not clear how well TimeX would perform on other tasks, such as anomaly detection or forecasting.

4. TimeX currently does not handle temporal prediction (i.e. a prediction task where prediction is done at every time-step), and it is unclear how it can be expanded to provide explanations for models designed for such tasks.

**Questions:**

1. In appendix, some of the explanations of the SeqComb-UV is visualized. The repeating patterns can all be seen to be happening around the same time points. Is TimeX capable of identifying patterns that are not necessarily aligned in absolute time between two patients?

2. Paper would be improved if there were additional experiments on different choices of $r$ presented, as at its current state, a reader unfamiliar with [46] will have a hard time deciphering the effect of different values of $r$.

3. Why is cosine similarity used in Equation 3? For SimCLR this would have made sense as the embedding of SimCLR is on the unit hyper-sphere and thus cosine similarity is an appropriate choice. But since the embeddings here are not constrained to be on the hyper-sphere I am uncertain why encoding through the notion of directionality would be helpful here.

4. Wouldn’t it be more appropriate to normalize $L_{MBC}$?

5. How does the differentiable attention masking presented in the appendix come into play for the model? Also this should be referring to section 4.2 instead of 4.1 as currently printed in the appendix.

6. Following the methods section, I am still uncertain as to why TIMEX is considered to be an in-hoc model and not a surrogate method. The paper would become easier to follow, if the authors can draw a parallel here and clear this uncertainty in the introduction.

7. The appendix currently provides an extensive list of experiments. One additional point to improve the already thorough analysis would be to visualize and see how much the explanations found differ for different underlying models tested (transformer, LSTM, CNN).

8. In the appendix the running time of the model is presented, but how does it compare to other baselines in your experiments?


Minor comments:
1. In section 3.1, $F^E$ should be defined as otherwise $F(z_i^E)$ would imply that $z_i^E$ and $z_i$ belong to the same space.
2. Line 213 has a typo: we [use] a direct-value masking procedure. ("use" is missing).
3. Specifics of how the transformer encoder-decoder was used should be added to appendix for reproducibility of results.

**Limitations:**

The paper does not contain any discussion on the potential societal impact of their work. As interpretability opens a broad avenue for discussion about this topic, I believe this section is essential to be added to the paper.

Additionally, the current limitations section is quite coarse. The authors can improve the paper by better expanding on the limitations of TimeX at its current state and the interesting directions for the future works.

---

> ### Author Rebuttal · Authors · 2023-08-09
>
> Thank you for your extensive comments, critiques, and praises for our work. We hope our response and experiments further convince you of TimeX’s novelty and effectiveness, and we kindly ask you to consider raising your score. Please reach out if there are additional questions.
> ### W1 and W2: Irregular time series
> We introduce an irregularly sampled version of SeqComb-MV, randomly dropping an average of 15% of time steps. This results in variable-length time series, addressing both W1 and W2. We then follow Zhang et al. [1] using a time series transformer with an irregular attention mask. To avoid direct interference with this mechanism, we train TimeX only with direct-value masking.
>
> **Table R2** shows the results of this experiment. We compare TimeX to Integrated Gradients (IG) performance because given the nuance of learning from irregularly-sampled datasets [1], most baselines do not apply to irregularly-sampled time series datasets without significant changes that are out-of-scope for our work. TimeX demonstrates superior performance in this setting, outperforming IG by an over 1.5x improvement in AUPRC.
> ### W3 and W4: TimeX on a wide range of tasks
> We agree that TimeX can in principle be applied to a wide range of tasks. We focus on classification for close comparison to evaluations in prior works. Still, to address your concerns, we demonstrate TimeX’s generalizability to diverse tasks by explaining a forecasting model. We use the ETTh1 dataset [2] and a vanilla forecasting transformer. To modify TimeX for forecasting, we first extract embeddings used for MBC by max pooling over hidden states of the decoder. Next, we used a revised LA loss that used MSE as the distance function between predictions rather than JS divergence.
>
> We show visualizations of two samples in **Figure R1**. Explanations are in the left column while forecasted time steps are in the right column. A few patterns emerge:
> 1. TimeX identifies late time steps as important for the forecast, an expected result for a dataset with small temporal variance.
> 2. **(a)**: the forecast is an increasing sequence, and TimeX identifies a region of an increasing sequence (time ~450-485). This suggests the model is using this increasing sequence as a reference for forecasting.
> 3. **(b)**: a sharp upward spike is forecasted at the beginning of the predicted window. Correspondingly, TimeX identifies a local minimum around time 260 and a sharp upward spike around time 475-490.
>
> TimeX extracts meaningful explanations from forecasting datasets. Regarding anomaly detection and temporal prediction, TimeX could be transferred by similarly modifying the training procedure. Given the nuances of these individual tasks, we leave this for future work.
> ### Q1: SeqComb-UV visualization
> The randomly-chosen samples for SeqComb-UV visualization have ground-truth explanations that fall in similar time points, but this is not true in general across our synthetic datasets. Each synthetic dataset contains different temporal dynamics that vary across time, and because of TimeX’s consistently-high performance, we are confident in its capability to identify important patterns in a wide variety of time series settings.
> ### Q2: Varying r parameter
> We conduct an experiment where we vary the $r$ parameter and measure explanation quality. We use the SeqComb-UV dataset and hold all hyperparameters constant while varying the $r$ parameter. The result is visualized in **Figure R3**. Low $r$ values lead to a drop in explainer performance with respect to AUPRC and AUP. Importantly, for $r$ values above 0.4, the explainer performance is stable, suggesting that TimeX is robust to choice of $r$ value.
> ### Q3: Cosine similarity
> In principle, Equation 3 can accept any distance function, but we choose cosine similarity because it is used in popular approaches such as SimCLR and InfoNCE [3,4].
> ### Q4: Normalizing $L_{MBC}$
> We do indeed normalize $L_{MBC}$ during training. We have made an appropriate amendment to Section 4.3 to fix this error.
> ### Q5: Differentiable attention masking
> We use the differentiable masking approach presented in Nguyen et al. [5] to ensure that we can learn the masks in TimeX in an end-to-end differentiable manner.
> ### Q6: In-hoc vs. surrogate method
> Thank you for pointing out this error. We have adopted the term “surrogate” to refer to our method. Please reference Point 2 in the general response.
> ### Q7: Explanations from different models
> We have generated **Figure R2** to show the explanations across different models on the FreqShapes dataset. Explanations are similar across models. TimeX and IG outputs appear similar, but TimeX has higher recall for important patterns, which is reflected in quantitative results.
> ### Q8: Runtime
> Please see general response Point 3.
> ### Minor Comments
> Thank you for pointing out several typos and errors in our submission. We have made these edits within the text.
> ### Limitations:
> In response to your comment, we will provide a more detailed limitations section and a societal impacts section. Our societal impacts section will read: “Time series data spans domains like finance, healthcare, energy, and transportation. Enhanced interpretability can bolster decision-making and trust. While it aids in identifying systemic biases for fairer, inclusive systems, caution is paramount. Misinterpretations or over-reliance on automated insights can backfire, highlighting the need for human and algorithmic collaboration.”
>
> **References**: [1] Zhang et al., “Graph-guided network for irregularly sampled multivariate time series”, ICLR 2022. [2] Zhou et al, “Informer: Beyond Efficient Transformer for Long Sequence Time-Series Forecasting”, AAAI 2021
> [3] Chen et al., “A Simple Framework for Contrastive Learning of Visual Representations”, ICML 2020.
> [4] van den Oord et al., “Representation Learning with Contrastive Predictive Coding”, arXiv 2018.
> [5] Nguyen et al., “Differentiable window for dynamic local attention”, ACL 2020.

---

> > ### Comment · Reviewer_Uw18 · 2023-08-19
> > **Response to the Rebuttal**
> >
> > I want to express my sincere appreciation for your comprehensive rebuttal to my review of your paper. Your responses clarified several aspects of the paper and addressed the concerns I had raised effectively.
> >
> > After carefully considering your responses and the additional information you provided to all the reviewers, I am convinced of the value and significance of your research. I believe that your paper makes a valuable contribution to the field. I commend your dedication to improving the paper and your willingness to engage in constructive dialogue. Your efforts have further strengthened my confidence in the quality and relevance of your work. For this reason I have increased my score to recommend accepting the paper for publication subject to including the additional experiments and explanations in the final version of the paper.
> > I have also increased my score for presentation (from 3 to 4) and confidence (from 3 to 4).
> >
> > Lastly, I would highly suggest including the synthetic data experiments and a sample jupyter notebook of end-to-end training and extracting the landmarks in your final published code for reproducibility.

---

### Author Rebuttal · Authors · 2023-08-09

Thank you to all the reviewers for thoughtful and insightful feedback! We are pleased that reviewers are excited about the novel contributions of our work. Reviewers remark that TimeX **“provide[s] interpretable explanations that are both faithful to the model’s predictions and informative about the underlying temporal patterns”** [Uw18]. Reviewers are also impressed with our strong evaluation results, noting that **“the evaluation is very convincing, using diverse datasets and many compared methods”** [9pJd] and that TimeX **“provides significant performance improvements in synthetic and real-world dataset[s]”** [Mf92]. Finally, reviewers find our work **“particularly relevant as high capacity models become more … accessible”** [wJLT], noting how TimeX **“shows to be effective in explaining a variety of time series models”** [Uw18] and **“can be generalized to other classification tasks and neural model architectures”** [7JHq]. We thank the reviewers for the strong praise of our work and contributions.

We now highlight a few important points raised by reviewers that warrant inclusion in the general response:
### Highlights of New Experiments
In response to reviewers comments, we have run 6 additional experiments and generated 4 additional plots. We have included these experimental details and results in individual rebuttals, with an experiment on computational efficiency detailed here in the overall response. We have also attached a single PDF page containing the four figures and two tables we generated in response to reviewer comments. We now briefly describe these experiments and the figures/tables referenced in each:
* **Runtime experiment** [Uw18,Mf92]: We show that TimeX is much faster at inference time than time series explainer baselines. Results are presented in **Response Table 1**. Please see Point 3 in the general response below.
* **Irregular time series** [Uw18]: We show that TimeX can achieve high explanation performance on an irregular time series synthetic dataset. The table is shown in **Response Table 2**.
* **Forecasting** [Uw18]: We demonstrate that TimeX can be used to explain forecasting tasks. A visualization is shown in **Response Figure 1**.
* **Visualizing explanations across model architectures** [Uw18]: We show a visualization of TimeX explanations across the transformer, CNN, and LSTM architecture on the FreqShapes dataset. This visualization is shown in **Response Figure 2**.
* **Varying $r$ parameter** [Uw18]: We find that TimeX is robust to choices of the $r$ parameter in the explanation loss. A visualization of these results is shown in **Response Figure 3**.
* **Applying IoU metric** [9pJd]: We add a new metric, Intersection-over-Union (IoU), that corroborates TimeX’s high explanation performance. The results are presented in the rebuttal to Reviewer 9pJd.
* **Landmark analysis** [wJLT]: We show that the landmarks learned by TimeX are high-quality and capture regions of the latent space with high similarity. Results are visualized in **Response Figure 4**.

### Note about in-hoc vs. surrogate explainers
We mistakenly referred to TimeX as an “in-hoc predictor” in-text. We appreciate Reviewer Uw18 pointing out this error. While the result of the model is an end-to-end differentiable time series explainer, TimeX is fundamentally a surrogate explainer method. We have appropriately removed the claims of TimeX being an in-hoc method and have clarified that it is a surrogate method in-text.

### Computational runtime analysis
We thank Reviewer Uw18 and Reviewer Mf92 for requesting more analysis of the computational resources required for TimeX against baseline explainers. Table X shows the training and inference time in seconds of TimeX versus two state-of-the-art time series-specific baselines, Dynamask and WinIT. We choose two real-world time series datasets, PAM and Epilepsy, which are of varying sizes. PAM contains 4266 training samples and 534 testing samples, each of 600 time steps in length. Epilepsy contains 8280 training samples and 2300 testing samples, each of 178 time steps in length. **Table 1** shows the time in seconds needed to train each explainer and to perform inference on the testing set.

TimeX is by-far the most efficient model at inference time for both datasets. This result is expected, as Dynamask and WinIT both require iterative procedures for each sample at inference time, while TimeX requires only a forward pass of $H^E$ at inference. Combining training and inference time, TimeX is the second-fastest on both datasets. However, WinIT and Dynamask times vary greatly between each dataset, with Dynamask as the fastest on PAM while WinIT the fastest on Epilepsy. WinIT scales poorly to samples with many time steps while Dynamask scales poorly to large testing sets. TimeX strikes a compromise between these extremes, scaling better than Dynamask to large numbers of testing samples while scaling better than WinIT to long time series.

### General Notes to Reviewers
Every figure and table reference starting with “R#” refers to figures and tables in the PDF attached to the general response; to make this clear, we bold figure names such as **Figure R1**. We abbreviate weaknesses by “W#” and questions by “Q#”. Despite being barred from submitting a revised manuscript, we assure the reviewers that edits have been incorporated into our revisions. We also use several common acronyms throughout the responses:
* XAI: Explainable artificial intelligence
* MBC: Model behavior consistency (Section 4.3)
* LA: Label alignment
* AUPRC, AUP, AUR: Metrics mentioned in Appendix C.4
We thank all reviewers again for their thoughtful commentary. We worked hard to improve our paper, and we sincerely hope the reviewers find our responses informative and helpful. If you feel the responses have not addressed your concerns to motivate increasing your score, we would love to hear what points of concern remain and how we can improve our work. Thank you again!

---

### Comment · Area_Chair_wu5B · 2023-08-15
**please engage in discussion (if you haven't already)**

Hello reviewers.

If you haven't already, please read the authors' response to your reviews *and to the other reviewers' reviews*, and discuss. The authors have provided reasonably detailed rebuttals so it would be great if you engage in discussion as soon as possible, especially to indicate if your opinion of the paper has changed or if you would like any sort of additional comments or clarifications.

Thanks, Area chair

---

### Decision · Program_Chairs · 2023-09-21

**Decision:**

Accept (spotlight)

**Comment:**

The authors provided very detailed responses to reviews to address concerns including running 6 additional experiments, and the reviewers' ratings unanimously leaned toward acceptance. I thus recommend accepting this paper as well. Especially as this paper addresses a rather broadly applicable problem of interpreting time series models with very promising results, I am further nominating this paper for a spotlight.